# Prediction and Control in Continual Reinforcement Learning

**Nishanth Anand** [*]
School of Computer Science
McGill University and Mila
nishanth.anand@mail.mcgill.ca

**Doina Precup**
School of Computer Science
McGill University, Mila, and Deepmind
dprecup@cs.mcgill.ca

## Abstract

Temporal difference (TD) learning is often used to update the estimate of the value function which is used by RL agents to extract useful policies. In this paper, we focus on value function estimation in continual reinforcement learning. We propose to decompose the value function into two components which update at different timescales: a *permanent* value function, which holds general knowledge that persists over time, and a *transient* value function, which allows quick adaptation to new situations. We establish theoretical results showing that our approach is well suited for continual learning and draw connections to the complementary learning systems (CLS) theory from neuroscience. Empirically, this approach improves performance significantly on both prediction and control problems.

## 1 Motivation

Deep reinforcement learning (RL) has achieved remarkable successes in complex tasks, e.g Go [40, 41] but in a narrow setting, where the task is well-defined and stable over time. In contrast, humans continually learn throughout their life and adapt to changes in the environment and in their goals. This ability to continually learn and adapt will be crucial for general AI agents who interact with people in order to help them accomplish tasks. One possible explanation for this ability in the natural world is the existence of complementary learning systems (CLS): one which *slowly* acquires structured knowledge and the second which *rapidly* learns specifics adapting to the current situation [26]. In contrast, in RL, learning is typically focused on how to optimize return, and the main estimated quantity is one value function (which then drives learning a policy). As a result, when changes occur in the environment, RL systems are faced with the *stability-plasticity dilemma* [9]: whether to forget past predictions in order to learn new estimates or to preserve old estimates, which may be useful again later, and compromise the accuracy on a new task.

The stability-plasticity dilemma in RL is agnostic to the type of function approximation used to estimate predictions, and it is not simply a side effect of the function approximator having small capacity. In fact, the problem exists even when the RL agent uses a table to represent the value function. To see this, consider policy evaluation on a sequence of tasks whose true value function is different due to different rewards and transition probabilities. A convergent temporal-difference (TD) algorithm [43] aggregates information from all value functions in the task distribution implicitly, thereby losing precision in the estimate of the value function for the current situation. A tracking TD learning algorithm [46] learns predictions for the current task, overwriting estimates of past tasks and re-learning from scratch for each new task, which can require a lot of data. Using a separate function approximator for each task, which is a common approach [23], requires detecting the identity of the current task to update accordingly, which can be challenging and limits the number of tasks that can be considered.

---

[*] corresponding author.

37th Conference on Neural Information Processing Systems (NeurIPS 2023).

As an alternative, we propose a CLS-inspired approach to the stability-plasticity dilemma which relies on maintaining two value function estimates: a *permanent* one, whose goal is to accumulate "baseline" knowledge from the entire distribution of information to which the agent is exposed over time, and a *transient* component, whose goal is to learn very quickly using information that is relevant to the current circumstances. This idea has been successfully explored in the context of RL and tree search in the game of Go [39], but the particular approach taken uses domain knowledge to construct features and update rules. Our approach is general, simple, online and model-free, and can be used in continual RL. Our method is also orthogonal to various advancements in the continual RL field, and therefore, it can be combined with other improvements. Our contributions are:

- A conceptual framework for defining permanent and transient value functions, which can be used to estimate value functions;
- RL algorithms for prediction and control using permanent and transient value functions;
- Theoretical results which help clarify the role of these two systems in the semi-continual (multi-task) RL setting; and
- Empirical case studies of the proposed approaches in simple gridworlds, Minigrid [11], JellyBeanWorld (JBW) [31], and MinAtar environments [51].

## 2   Background

Let $\mathcal{S}$ be the set of possible states and $\mathcal{A}$ the set of actions. At each timestep $t$, the agent takes action $A_t \in \mathcal{A}$ in state $S_t$ according to its (stochastic) policy $\pi : \mathcal{S} \to Dist(\mathcal{A})$. As a consequence, the agent receives reward $R_{t+1}$ and transitions to a new state $S_{t+1}$. Let $G_t = \sum_{k=t}^{\infty} \gamma^{k-t} R_{k+1}$ be the discounted return obtained by following $\pi$ from step $t$ onward, with $0 < \gamma < 1$ the discount factor. Let $v_\pi(s) = \mathbb{E}_\pi[G_t | S_t = s]$. This quantity can be estimated with a function approximator parameterized by $\mathbf{w}$, for example using TD learning [43]:

$$\mathbf{w}_{t+1} \leftarrow \mathbf{w}_t + \alpha_t \delta_t \nabla_{\mathbf{w}} v_{\mathbf{w}}(S_t), \tag{1}$$

where $\alpha_t$ is the learning rate at time $t$, $\delta_t = R_{t+1} + \gamma v_{\mathbf{w}}(S_{t+1}) - v_{\mathbf{w}}(S_t)$ is the TD error (see Sutton and Barto [45] for details).

In the control problem, the agent's goal is to find a policy, $\pi^*$, that maximizes expected returns. This can be achieved by estimating the optimal action-value function, $q^* = max_\pi q_\pi$, where $q_\pi = \mathbb{E}_\pi[G_t | S_t = s, A_t = a]$, e.g. using Q-learning [50].

"Continual RL" is closely related to transfer learning [48], meta learning [37, 15], multi-task learning [10], and curriculum learning [35]. A thorough review distinguishing these flavours can be found in Khetarpal et al. [24]. We use Continual RL to refer to a setup in which the agent's environment (i.e. the reward or transition dynamics) changes over time. We use the term *semi-continual RL* for the case when the agent can observe task boundaries (this is similar to the multi-task setting, but does not make assumptions about the task distribution or task IDs). In continual RL, a unique optimal value function may not exist, so the agent has to continually *track* the value function in order to learn useful policies [46, 2].

Our work focuses on the idea of two learning systems: one learning the "gist" of the task distribution, slowly, the other which is used for tracking can quickly adapt to current circumstances. We instantiate this idea in the context of learning value functions, both for prediction and control, but the idea can be applied broadly (for e.g. policies, GVFs). The concept of decomposing the value function into permanent and transient components was first introduced in model-based RL [39], for the single task of learning to play Go, and using samples drawn from the model to train the transient value function. In contrast, we focus on general continual RL, with samples coming from the environment.

In continual RL, most of the literature is focused on supplementing neural network approaches with better tools, such as designing new optimizers to update weights [44, 19, 12, 4, 14], building new architectures [21, 32], using experience replay to prevent forgetting [34, 36, 22, 8], promoting plasticity explicitly [28, 29], or using regularization techniques from continual supervised learning [25, 20, 30, 27]. Our method is agnostic to the nature of the function approximator used, and therefore, complementary to these advancements (and could in fact be combined with any of these methods).

Our approach has parallels with fast and slow methods [13, 7] which use neural networks as the main learning system and experience replay as a secondary system. Another similar flavor is to use the task

distribution to learn a good starting point, as in distillation-based approaches [49, 18] or MaxQInit [1]. But the latter are multi-task RL methods and their use in continual RL is limited.

## 3 Defining Permanent and Transient Value Functions

Inspired by CLS theory, we decompose the value function estimated by the agent into two, independently parameterized components: one which slowly acquires general knowledge, the other which quickly learns local nuances but then forgets. We call these components *permanent value function*, $V_\theta^{(P)}$ and *transient value function*, $V_\mathbf{w}^{(T)}$. The overall value function is computed additively:

$$V^{(PT)}(s) = V_\theta^{(P)}(s) + V_\mathbf{w}^{(T)}(s), \tag{2}$$

A similar decomposition can be done to the action-value function:

$$Q^{(PT)}(s,a) = Q_\theta^{(P)}(s,a) + Q_\mathbf{w}^{(T)}(s,a). \tag{3}$$

Other approaches to combining the two components are possible but additive decomposition is the simplest.

### 3.1 Permanent Value Function

Similar to the role of the neocortex in the brain, the permanent value function should capture general structure in the value functions from the tasks that the agent has seen in its lifetime. It should also provide good baseline predictions for any task that the agent faces, in order to allow it to learn good estimates with fewer updates. To achieve this in a scenario where the agent goes through a sequence of tasks and knows when the task changes, we might store all the states visited during the current task in a buffer and using them when the task finishes, to update the permanent value function as follows:

$$\theta_{k+1} \leftarrow \theta_k + \overline{\alpha}_k(V^{(PT)}(S_k) - V_\theta^{(P)}(S_k))\nabla_\theta V_\theta^{(P)}(S_k), \tag{4}$$

where $\overline{\alpha}$ is the learning rate and $k$ is the iteration index of the buffer. The permanent estimate takes a *small* step towards the new estimate of the value function in the above update rule: bootstrapping at a slower timescale. We analyze this idea further in Sec. 4.

### 3.2 Transient Value Function

The transient value function should compute corrections on top of the estimates provided by the permanent value function, in order to approximate more closely the true value function of the current task. We implement this intuition by updating the parameters of transient value function as:

$$\mathbf{w}_{t+1} \leftarrow \mathbf{w}_t + \alpha_t \delta_t \nabla_\mathbf{w} V_\mathbf{w}^{(T)}(S_t), \tag{5}$$

where $\alpha$ is the learning rate, $\delta_t = R_{t+1} + \gamma V^{(PT)}(S_{t+1}) - V^{(PT)}(S_t)$ and $\nabla_\mathbf{w} V_\mathbf{w}^{(T)}(S_t)$ is the gradient of transient value function at time $t$. Eq. (5) is a semi-gradient update rule, in the same sense used to describe TD-learning, as the gradient is with respect to the current estimate only and not the target. Since the permanent and the transient value functions are independently parameterized, the semi-gradient ends up being just $\nabla_\mathbf{w} V_\mathbf{w}^{(T)}(S_t)$. This results in learning only a part of the value function that is not captured by the permanent value function. The transient value function is reset or decayed after transferring the new information into the permanent value function (e.g. at the moment when the task changes), similar to the functioning of the hippocampus [26, 33].

### 3.3 Learning Algorithm

We can easily turn the updates above into algorithms; a blueprint for the prediction case is shown in Algorithm 1, and the control version is in Appendix 3. The initialization and resets are done appropriately based on the function approximation used.

In the above algorithm, we assumed that the agent can observe task boundaries, though not any additional information (like a task id). We made this assumption to facilitate the theoretical analysis presented in the next section. A fully continual version of the algorithms is presented and evaluated empirically in Sec. 6.

Note that this algorithm is a strict generalization of TD learning: instead of setting $\mathbf{w}$ to $0$ when the task changes, if we were to match the difference between the updated permanent value function and the transient value function instead, we would obtain exactly the TD learning predictions. Another way to get TD is by initializing $\theta$ to match the initialization of the TD learning algorithm and updating $\mathbf{w}$ only without resetting. We explore this connection formally in the next section.

---

**Algorithm 1** PT-TD learning (Prediction)

---
1: Initialize: Buffer $\mathcal{B}$, $\theta$, $\mathbf{w}$
2: **for** $t : 0 \rightarrow \infty$ **do**
3:     Store $S_t$ in $\mathcal{B}$
4:     Observe $R_{t+1}, S_{t+1}$
5:     Update $\mathbf{w}$ using Eq. (5)
6:     **if** Task Ends **then**
7:         Update $\theta$ using $\mathcal{B}$ and Eq. (4)
8:         Reset transient value function, $\mathbf{w}$
9:         Reset $\mathcal{B}$
10:     **end if**
11: **end for**

---

## 4 Theoretical Results

In this section, we provide some theoretical results aiming to understand the update rules of permanent and transient value function separately, and then we study the semi-continual RL setting (because the fully continual RL setting is poorly understood in general from a theoretical point of view, and even the tools and basic setup are somewhat contentious). All our results are for the prediction problem with tabular value function representation; We expect that prediction results with linear function approximation can also be obtained. We present theorems in the main paper and include proofs in appendices B. The appendix also contains analytical equations describing the evolution of the parameters based on these algorithms.

### 4.1 Transient Value Function Results

We provide an understanding of what the transient value function does in the first set of results (Theorems 1-4). We carry out the analysis by fixing $V^{(P)}$ and only updating $V^{(T)}$ using samples from one particular task. We use $V_t^{(TD)}$ and $V_t^{(PT)}$ to denote the value function estimates at time $t$ learned using TD learning and Alg. 1 respectively.

The first result establishes a relationship between the estimates learnt using TD and the transient value function update rule in Eq.(5).

**Theorem 1.** *If $V_0^{(TD)} = V^{(P)}$, $V_0^{(T)} = 0$ and the two algorithms train on the same samples using the same learning rates, then $\forall t, V_t^{(PT)} = V_t^{(TD)}$.*

The theorem shows that the estimates learnt by our approach match those learnt by TD-learning in a special case, proving that our algorithm is a strict generalization of TD-learning.

The next two theorems together shows that the transient value function reduces prediction error. First we show the contraction property of the transient value function Bellman operator.

**Theorem 2.** *The expected target in the transient value function update rule is a contraction-mapping Bellman operator: $\mathcal{T}^{(T)}V^{(T)} = r_\pi + \gamma \mathcal{P}_\pi V^{(P)} - V^{(P)} + \gamma \mathcal{P}_\pi V^{(T)}$, where $r_\pi$ is expected one-step reward for each state written in vector form and $\mathcal{P}_\pi$ is the transition matrix for policy $\pi$.*

We get the Bellman operator by writing the expected target in matrix form. We now characterize the fixed point of the transient value function operator in the next theorem.

**Theorem 3.** *The unique fixed point of the transient operator is $(I - \gamma \mathcal{P}_\pi)^{-1} r_\pi - V^{(P)}$.*

The theorem confirms the intuition that the transient value function learns only the part of the value function that is not captured in the permanent value function, while the agent still learns the overall value function: $V^{(PT)} = V^{(P)} + V^{(T)} = V^{(P)} + (I - \gamma \mathcal{P}_\pi)^{-1} r_\pi - V^{(P)} = v_\pi$.

The next theorem relates the fixed point of the transient value function to the value function of a different Markov Reward Process (MRP)

**Theorem 4.** *Let $\mathcal{M}$ be an MDP in which the agent is performing transient value function updates for $\pi$. Define a Markov Reward Process (MRP), $\widetilde{\mathcal{M}}$, with state space $\widetilde{\mathcal{S}} : \{(S_t, A_t, S_{t+1}), \forall t\}$, transition dynamics $\widetilde{\mathcal{P}}(x'|x, a') = \mathcal{P}(s''|s', a')$, $\forall x, x' \in \widetilde{\mathcal{S}}, \forall s', s'' \in \mathcal{S}, \forall a' \in \mathcal{A}$ and reward function $\widetilde{\mathcal{R}}(x, a') = \mathcal{R}(s, a) + \gamma V^{(P)}(s') - V^{(P)}(s)$. Then, the fixed point of Eq.(5) is the value function of $\pi$ in $\widetilde{\mathcal{M}}$.*

The proof follows by comparing the value function of the modified MRP with the fixed point of transient value function. The theorem identifies the structure of the fixed point of the transient value function in the form of the true value function of a different MRP. This result can be used to design an objective for the permanent value function updates; For instance, to reduce the variance of transient value function updates.

## 4.2 Permanent Value Function Results

We focus on permanent value function in the next two theorems. First, we characterize the fixed point of the permanent value function. Then, we connect the fixed point of the permanent value function to the jumpstart objective [48, 1], which is defined as the initial performance of the agent in the new task before collecting any data. The latter is used to measure the effectiveness of transfer in a zero-shot learning setup, and therefore, is crucial to measure generalization under non-stationarity. We make an assumption on the task distribution for the following proofs:

**Assumption 1.** Let MDP $\mathcal{M}_\tau = (\mathcal{S}, \mathcal{A}, \mathcal{R}_\tau, \mathcal{P}_\tau, \gamma)$ denote task $\tau$. Then, $v_\tau$ is the value function of task $\tau$ and let $\mathbb{E}_\tau$ denote the expectation with respect to the task distribution. We assume there are $N$ tasks and task $\tau$ is i.i.d. sampled according to $p_\tau$.

**Theorem 5.** *Following Theorem 2, under assumption 1 and Robbins-Monro step-size conditions, the sequence of updates computed by Eq.(4) contracts to a unique fixed point $\mathbb{E}_\tau[v_\tau]$.*

We use the approach described in Example 4.3 from Bertsekas and Tsitsiklis [5] to prove the result. Details are given in Appendix B.2.1

**Theorem 6.** *The fixed point of the permanent value function optimizes the jumpstart objective, $J = \min_{u \in \mathbb{R}^{|\mathcal{S}|}} \frac{1}{2}\mathbb{E}_\tau[\|u - v_\tau\|_2^2]$.*

## 4.3 Semi-Continual RL Results

We present two important results in this section which prove the usefulness of our approach in value function retention, for past tasks, and fast adaptability on new tasks. For the proofs, we assume that at time $t$, both $V_t^{(TD)}$ and $V_t^{(PT)}$ have converged to the true value function of task $i$ and the agent gets samples from task $\tau$ from timestep $t + 1$ onward.

The first result proves that the TD estimates forget the past after seeing enough samples from a new task, while our method retains this knowledge through $V^{(P)}$.

**Theorem 7.** *Under assumption 1, there exists $k_0$ such that $\forall k \geq k_0$, $\mathbb{E}_\tau\left[\left\|V_{t+k}^{(TD)} - v_i\right\|_2^2\right] > \mathbb{E}_\tau\left[\left\|V^{(P)} - v_i\right\|_2^2\right], \forall i$.*

The proof uses the sample complexity bound from Bhandari et al. [6] (Sec. 6.3, theorem 1). Details are in Appendix B.3.1.

Next, we prove that our method has a tighter upper bound on errors compared with the TD learning algorithm in expectation.

**Theorem 8.** *Under assumption 1, there exists $k_0$ such that $\forall k \leq k_0$, $\mathbb{E}_\tau\left[\left\|v_\tau - V_{t+k}^{(PT)}\right\|_2^2\right]$ has a tighter upper bound compared with $\mathbb{E}_\tau\left[\left\|v_\tau - V_{t+k}^{(TD)}\right\|_2^2\right], \forall i$.*

The proof uses the sample complexity bound from Bhandari et al. [6] (Sec. 6.3, theorem 1) to get the desired result as detailed in Appendix B.3.2. As $k \to \infty$, the estimates of both the algorithms converge to the true value function and their upper bounds collapse to 0. This result doesn't guarantee low error at each timestep, but it provides an insight on why our algorithm has low error.

To further understand the behaviour of our algorithm on a new task, we derive analytical expressions for mean squared error similar to Singh and Dayan [42] for both our algorithm and TD-learning algorithm (see Appendix B.3.3 and B.3.4). Then, we use them to calculate the mean squared error empirically on a toy problem. We observe low error for our algorithm at every timestep for various task distributions (details in Appendix C.2).

# 5 Experiments: Semi-Continual Reinforcement Learning

We conducted experiments in both the prediction and the control settings on a range of problems. Although the theoretical results handle the prediction setting, we performed control experiments to demonstrate the broad applicability of our approach. In this section, we assume that the agent is faced with a sequence of tasks in which the transition dynamics are the same but the reward function changes. The agent knows when the task has changed. In the sections below, we focus on the effectiveness of our approach. Additional experiments providing other insights (effect of network capacity, sensitivity to hyperparameters, and effect of task distribution) are included in the Appendix (Sec. C.2, C.3, and C.4).

**Baselines:** Because our approach is built on top of TD-learning and Q-learning, we primarily use the usual versions of TD learning and Q-learning as baselines, which can be viewed as emphasizing stability. We also include a version of these algorithms in which weights are reset when the task switches, which creates significant plasticity. We include more baselines for the continual RL experiments, which are described in Sec. 6. The results for all algorithms are averaged across 30 seeds and shaded regions in the figures indicate 90% confidence intervals[2]. We use the same procedure to optimize hyperparameters for all algorithms, detailed in Appendix (Sec. C.4).

## 5.1 Prediction experiments

We ran five experiments on 3 types of navigation tasks, with various function approximators. The policy being evaluated was uniformly random in all experiments. We used root mean squared value error (RMSVE) as a performance measure on the online task and the expected mean squared error (MSE) on the other tasks.

**Discrete grid:** is a 5x5 environment depicted in Fig. 1a, which has four goal states (one in each corner). The agent starts in the center and chooses an action (up, down, left or right). Each action moves the agent one step along the corresponding direction. Rewards are 0 for all transitions except those leading into a goal state. The goal rewards change as detailed in Table C.1, so the value function also changes as shown in Fig. C.1. For the linear setting, each state is represented using 10 features: five for the row and five for the column. The permanent and transient value functions use the same features. For the deep RL experiment, instead of receiving state information, the agent receives 48x48 RGB images. The agent's location is indicated in red, and goal states are colored in green

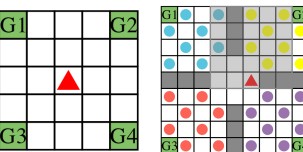

(a) Discrete grid.     (b) Minigrid.

Figure 1: Environments (PE).

as shown in Fig. 1a. Goal rewards are multiplied by 10 to diminish the noise due to the random initialization of the value function. For the tabular and linear setting, we run 500 episodes and change the goal rewards after every 50 episodes. For the deep RL experiments, we run 1000 episodes and goal rewards are switched every 100 episodes. The discount factor is 0.9 in all cases.

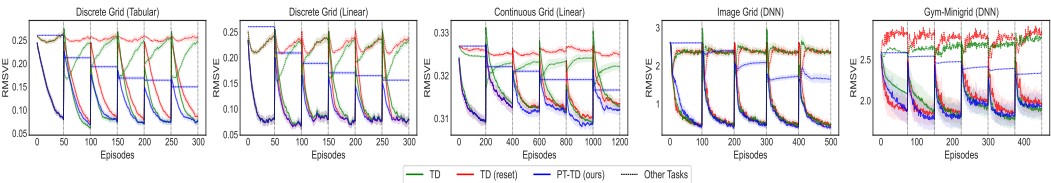

Figure 2: Prediction results on various problems. Solid lines represent RMSVE on the current task and dotted lines represent MSE on other tasks. Black dotted vertical lines indicate task boundaries.

**Continuous grid:** is a continuous version of the previous task. The state space is the square $[0, 1] \times [0, 1]$, with the starting state sampled uniformly from $[0.45, 0.55] \times [0.45, 0.55]$. The regions within $0.1$ units (in norm-1) from the corners are goal states. The actions change the state by $0.1$ units along the corresponding direction. To make the transitions stochastic, we add noise to the new state,

---

[2]We report standard distribution confidence intervals with a z-score of 1.645. We use $\mu \pm z\frac{\sigma}{\sqrt{n}}$ to represent the shaded region.

sampled uniformly from $[-0.01, 0.01] \times [-0.01, 0.01]$. Rewards are positive only for transitions entering a goal region. The goal reward for each task is given in Table C.1. In the linear function approximation setting we convert the agent's state into a feature vector using the radial basis function as described in [45] and Appendix C.1.1. We run 2000 episodes, changing goal rewards every 200 episodes. For evaluation, we sample 225 evenly spaced points in the grid and estimate their true values by averaging Monte Carlo returns from 500 episodes across 10 seeds for each task. Here, $\gamma = 0.99$.

**Minigrid:** We used the four rooms environment shown in Fig. 1b. Each room contains one type of item and each item type has a different reward and color. Goal states are located in all corners. The episode terminates when the agent reaches a goal state. The agent starts in the region between cells (3,3) and (6,6) and can move forward, turn left, or turn right. The agent perceives a 5x5 (partial) view in front of it, encoded one-hot. The rewards for picking items change as described in Table C.2 and the discount factor is 0.9. We run 750 episodes and change rewards every 75 episodes.

We provide the training and architecture details for the two deep RL prediction experiments in Appendix (Sec. C.1.1 and C.1.2) due to space constraints.

**Observations:** The results are shown in Fig. 2. Solid lines indicate performance on the task currently experienced by the agent (i.e., the *online* task) and the dotted lines indicate the performance on the other tasks. The reset variant of TD-learning performs poorly because all learning progress is lost when the weights are reset, leading to the need for many samples after every task change. TD-learning also has high online errors immediately after a task switch, as its predictions are initially biased towards the past task. As the value function is updated using the online interaction data, the error on the current task reduces and the error on other tasks rises, highlighting the stability-plasticity problem when using a single value function estimate. Our algorithm works well on both performance measures. The permanent value function learns common information across all tasks, resulting in lower errors on the second performance measure throughout the experiment. Also, due to the good starting point provided by the permanent value function, the transient component requires little data to make adjustments, resulting in fast adaptation on the new task. In our method, the permanent component provides stability and the transient component provides plasticity, resulting in a good balance between the two. We also plotted the heatmap of the value estimates, shown in Fig. C.3, to understand the evolution of the predictions for TD-learning and our method which confirms the observations made so far.

## 5.2 Control experiments

To test our approach as a complement to Q-learning 3, we conducted a tabular gridworld experiment and a Minigrid experiment using deep neural networks.

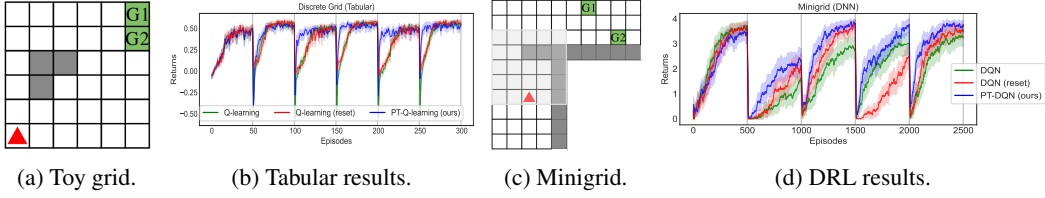

(a) Toy grid.   (b) Tabular results.   (c) Minigrid.   (d) DRL results.

Figure 3: The environments and the corresponding results in the control setting. (b) Tabular results: episodic returns are plotted. (d) Deep RL results: returns smoothened across 10 episodes is plotted.

**Tabular experiment:** We use the 6x6 grid shown in Fig. 3a, with two goals located in the top right corner. The agent starts in the bottom left corner and can choose to go up, down, left, or right. The action transitions the agent to the next state along the corresponding direction if the space is empty and keeps it in the same state if the transition would end up into a wall or obstacle. The action has the intended effect 90% of the time; otherwise, it is swapped with one of the perpendicular actions. The agent receives a reward of +1 and -1 for reaching the two goal states respectively, and the rewards swap periodically, creating two distinct tasks. The discount factor is 0.95. We run 500 episodes and change the task every 50 episodes.

**Minigrid experiment:** We use the two-room environment shown in Fig. 3c. The agent starts in the bottom row of the bottom room and there are two goal states located in the top room. The goal

rewards are +5 and 0 respectively, and they flip when the task changes. We run 2500 episodes, alternating tasks every 500 episodes. The other details are as described in the previous section. Further details on the training and architecture are described in Appendix (Sec. C.1.3).

**Observations:** The results are shown in Fig. 3. The reset variant of Q-learning has to learn a good policy from scratch whenever the task changes, therefore, it requires many samples. Since, the goal rewards are flipped from one task to another, Q-learning also requires many samples to re-adjust the value function estimates. In our method, since the permanent component mixes action-values across tasks, the resulting bias is favorable, and makes it fairly easy to learn the corrections required to obtain overall good estimates. So, the transient component requires comparatively few samples to find a good policy when the task changes.

# 6 Continual Reinforcement Learning

So far, we assumed that the agent observes task boundaries; This information is usually not available in continual RL. One approach that has been explored in the literature is to have the agent infer, or learn to infer, the task boundaries through observations (e.g. see [17, 23]). However, these methods do not work well when task boundaries are not detected correctly, or simply do not exist. We will now build on the previous algorithm we proposed to obtain a general version, shown in Alg. 2, which is suitable for continual RL.

Our approach maintains the permanent and transient components, but updates them continually, instead of waiting for task changes. The permanent component is updated every $k$ steps (or episodes) using the samples stored in the buffer $\mathcal{B}$. And, instead of resetting the transient component, it is now decayed by a factor $\lambda \in [0, 1]$. Here, $k$ and $\lambda$ are both hyperparameters. When

---

**Algorithm 2** PT-Q-learning (CRL)

1: Initialize: Buffer $\mathcal{B}$, $\theta$, $\mathbf{w}$, $k$, $\lambda$
2: **for** $t : 0 \to \infty$ **do**
3:     Take action $A_t$
4:     Store $S_t$, $A_t$ in $\mathcal{B}$
5:     Observe $R_{t+1}$, $S_{t+1}$
6:     Update $\mathbf{w}_t$ using Eq. (8)
7:     **if** $mod(t, k) == 0$ **then**
8:         Update $\theta$ using $\mathcal{B}$ and Eq. (7)
9:         $\mathbf{w}_{t+1} \leftarrow \lambda \mathbf{w}_{t+1}$
10:       Reset $\mathcal{B}$
11:     **end if**
12: **end for**

---

$\lambda = 0$ and $k$ the duration between task switches, we obtain the previously presented algorithm as a special case. In general, $k$ and $\lambda$ control the stability-plasticity trade off. Lower $\lambda$ means that the transient value function forgets more, thus retaining more plasticity. Lower $k$ introduces more plasticity in the permanent value function, as its updates are based on fewer and more recent samples. The algorithm still retains the idea of consolidating information from the transient into the permanent value function.

**The effect of $k$ and $\lambda$:** To illustrate the impact of the choice of hyperparameters, we use the tabular task shown in Fig. 3a. The details of the experiment are exactly as before, but the agent does not know the task boundary. We run the experiment for several combinations of $k$ and $\lambda$ for the best learning rate. The results are shown in Fig. 4. As expected, we observe that the hyperparameters $k$ and $\lambda$ have co-dependent effects to a certain degree. For small values of $k$, large values of $\lambda$ yield better performance, because the variance in the permanent value function is mitigated, and the transient value function in fact is more stable. For large values of $k$, the transient value function receives enough updates before the permanent value function is trained, to attain low prediction error. Therefore, the updates to the permanent value function are effective, and the transient predictions can be decayed more aggressively. More surprisingly, if the frequency of the permanent updates is significantly higher or significantly lower than the frequency of task changes, the overall performance is marginally affected, but remains better than for Q-learning (indicated using black dotted line) for most $k$ and $\lambda$ values. In fact, for each $k$, there is at least one $\lambda$ value for which the performance of the proposed algorithm exceeds the Q-learning baseline. So, in practice, we could fix one of these hyperparameters beforehand and tune only the second. We suspect that the two parameters could be unified into a single one, but this is left for future work.

## 6.1 Experiments

We test our algorithm on two domains: JellyBeanWorld (JBW) [31] and MinAtar [51]. We chose these domains because of the relatively lighter computational requirements, which still allow us to

sweep a range of hyperparameters and to report statistical results averaged over 30 seeds. We include architecture and further training details in Appendix (Sec. C.1.5, C.1.6).

**JellyBeanWorld** is a benchmark to test continual RL algorithms [31]. The environment is a two-dimensional infinite grid with three types of items: blue, red, and green, which carry different rewards for different tasks as detailed in Sec. C.1.5. In addition to reward non-stationarity, the environment has spatial non-stationarity as shown in Fig. 5a. The agent observes an egocentric 11x11 RGB image and chooses one of the four usual navigation actions. We run 2.1M timesteps, with the task changing every 150k steps. The discount factor is 0.9.

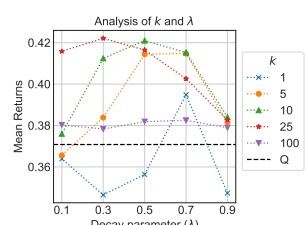

Figure 4: $\lambda$ and $k$ analysis.

**MinAtar** is a standard benchmark for testing single-task RL algorithms [51]. We use the breakout, freeway, and space invaders environments. We run for 3.5M steps and setup a continual RL problem, by randomly picking one of these tasks every 500k steps. We standardize the observations across tasks by padding with 0s when needed and we make all the actions available to the agent. This problem has non-stationarity in transitions, rewards and observations making it quite challenging. The discount factor is 0.99. More details are included in Appendix (Sec. C.1.6).

**Baselines:** We compare our method with DQN, two DQN variants, and a uniformly random policy. For the first baseline, we use a large experience replay buffer, an approach which was proposed recently as a viable option for continual RL [8]. The second baseline, DQN with multiple heads, uses a common trunk to learn features and separate heads to learn Q-values for the three tasks. This requires knowing the identity of the current task to select the appropriate head. We use this baseline because it is a standard approach for multi-task RL, and intuitively, it provides an upper bound on the performance. This approach is not suitable if the number of tasks is very high, or if the problem is truly continual and not multi-task. For our approach, we use half the number of parameters as that of DQN for both permanent and transient value networks to ensure the total number of parameters across all baselines are same — hence the name PT-DQN-0.5x.[3]

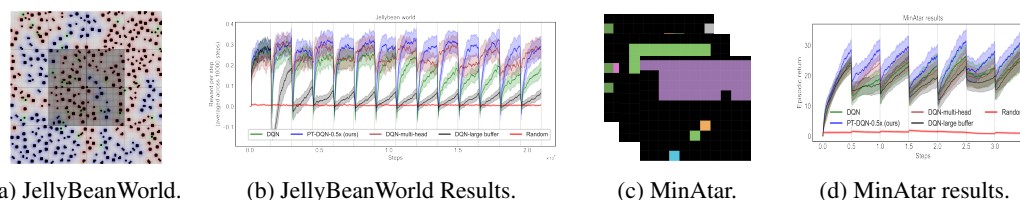

(a) JellyBeanWorld.  (b) JellyBeanWorld Results.  (c) MinAtar.  (d) MinAtar results.

Figure 5: (a) JellyBeanWorld task. (b) Results: JellyBeanWorld. (c) MinAtar tasks. (d) Results: MinAtar.

**Results** are presented in Fig. 5 and the pseudocode of PT-DQN is included in Appendix C.1.4. For JBW experiment, we plot the reward obtained per timestep over a 10k step window (reward rate) as a function of time. For MinAtar, we plot the return averaged over the past 100 episodes. Our method, PT-DQN-0.5x, performs better than all the baselines. The performance boost is large in the JBW experiment, where the rewards are completely flipped, and still favourable in the MinAtar experiment. DQN's performance drops over time when it needs to learn conflicting Q-values, as seen in the JBW results. DQN also needs a large number of samples to overcome the bias in its estimates when the task changes. Surprisingly, multi-headed DQN performs poorly compared to our method, despite having the additional task boundary information. The weights of the trunk are shared, and they vary continuously over time. Therefore, the weights at the head become stale compared to the features, which degrades performance. These findings may be different if the trunk were pre-trained and then fixed, with only the heads being fine-tuned, or if the networks were much bigger (which is however not warranted for the scale of our experiment). Augmenting DQN with a large experience replay is detrimental for general non-stationary problem, based on our experiments. This is because the samples stored in the replay buffer become outdated, so using these samples hampers the agent's ability to track environment changes.

---

[3]Strictly speaking, DQN-based baselines use more parameters compared with ours because their target networks are 2x larger than the target network of the transient value function.

# 7 Discussion

Designing agents that can learn continually, from a single stream of experience, is a crucial challenge in RL. In this paper, we took a step in this direction by leveraging intuitions from complementary learning systems, and decomposing the value function into permanent and transient components. The permanent component is updated slowly to learn general estimates, while the transient component computes corrections quickly to adapt these estimates to changes. We presented versions adapted to both semi and fully continual problems, and showed empirically that this idea can be useful. The theoretical results provide some intuition on the convergence and speed of our approach.

The non-stationarity in our experiments is much more challenging, including arbitrary reward changes, as well as dynamics and observation changes, compared to other works, which use specific structure, or rely on a curriculum of tasks (thereby limiting non-stationarity). Our approach provided improvements in performance compared to the natural TD-learning and Q-learning baselines.

Our approach performs fast and slow interplay at various levels. The permanent value function is updated at a slower timescale (every $k$ steps or at the end of the task); The transient value function is updated at a faster timescale (every timestep). The permanent value function is updated to capture some part of the value function from the agent's entire experience (generalization), which is a slow process; The transient value function adapts the estimates to a specific situation, which is a fast process. The permanent value function is updated using a smaller learning rate (slow), but the transient value function is updated using a larger learning rate (fast) [3].

## 7.1 Limitations and Future Work

While we focused on illustrating the idea of permanent and transient components with value-based algorithms with discrete action spaces, the approach is general and could in principle be used with policy gradient approaches. One could use the proposed value function estimates to compute the gradient of the policy (represented as a separate approximator) in actor-critic methods. Another possibility is to have a "permanent" policy, which is corrected by a transient component to adapt to the task at hand. These directions would be interesting to explore. Our idea can be extended to learning General Value Functions [47] too.

Our theoretical analysis is limited to piece-wise non-stationarity, where the environment is stable for some period of time. An interesting direction would be to extend our approach to other types of non-stationarity, such as continuous wear and tear of robotic parts.

Another important future direction is to consider distinct feature spaces for the permanent and transient components, as done in Silver et al. [38]. This would likely increase the ability to trade off stability and plasticity. Another direction to better control the stability and plasticity trade-off is by learning the frequency parameter $k$ and the decay parameter $\lambda$ using meta objectives [16]; We suspect there's a relation between the two parameters that can unify them into a single one.

Our approach can also be used to bridge offline pre-training and online fine-tuning of RL agents. The permanent value function can be learnt using offline data and these estimates can be adjusted by computing corrections using the transient value function using online interaction data.

## Acknowledgments and Disclosure of Funding

We are grateful to NSERC, CIFAR, and Microsoft Research grants for funding this research. We thank Jalaj Bhandari, Maxime Webartha, Matthew Reimer, Blake Richards, Emmanuel Bengio, Harsh Satija, Sumana Basu, Chen Sun, and many other colleagues at Mila for the helpful discussions; Will Dabney, Harm Van Seijen, Rich Sutton, Razvan Pascanu, David Abel, James McClelland and all of the 2023 Barbados workshop participants for their feedback on the project; the anonymous reviewers for the constructive feedback on the final draft. This research was enabled in part by support provided by Calcul Quebec and the Digital Research Alliance of Canada.

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

# Appendix

## A    Control algorithm

The action-value function can be decomposed into two components as:

$$Q^{(PT)}(s,a) = Q_\theta^{(P)}(s,a) + Q_\mathbf{w}^{(T)}(s,a), \tag{6}$$

The updates for permanent value function are:

$$\theta_{k+1} \leftarrow \theta_k + \overline{\alpha}_k(\hat{Q}(S_k, A_k) - Q_\theta^{(P)}(S_k, A_k))\nabla_\theta Q_\theta^{(P)}(S_k, A_k). \tag{7}$$

where $\overline{\alpha}$ is the learning rate. An update rule similar to Q-learning can be used for the transient value function:

$$\mathbf{w}_{t+1} \leftarrow \mathbf{w}_t + \alpha_t \delta_t \nabla_\mathbf{w} Q_\mathbf{w}^{(T)}(S_t, A_t), \tag{8}$$

where $\delta_t = R_{t+1} + \gamma \max_a Q^{(PT)}(S_{t+1}, a) - Q^{(PT)}(S_t, A_t)$ is the TD error.

---

**Algorithm 3** PT-Q-learning (Control)

---

1: Initialize: Buffer $\mathcal{B}$, $\theta$, $\mathbf{w}$
2: **for** $t : 0 \rightarrow \infty$ **do**
3:     Take action $A_t$
4:     Store $S_t$, $A_t$ in $\mathcal{B}$
5:     Observe $R_{t+1}$, $S_{t+1}$
6:     Update $\mathbf{w}$ using Eq. (8)
7:     **if** Task Ends **then**
8:         Update $\theta$ using $\mathcal{B}$ and Eq. (7)
9:         Reset transient value function, $\mathbf{w}$
10:         Reset $\mathcal{B}$
11:     **end if**
12: **end for**

---

## B    Proofs

### B.1    Transient Value Function Results

We carry out the analysis by fixing $V^{(P)}$ and only updating $V^{(T)}$ using samples from one particular task. We use $V_t^{(TD)}$ and $V_t^{(PT)}$ to denote the value function estimates at time $t$ learned using TD learning and Alg. 1 respectively.

**Theorem B.1.1.** *If $V_0^{(TD)} = V^{(P)}$, $V_0^{(T)} = 0$ and the two algorithms train on the same samples using the same learning rates, then $\forall t, V_t^{(PT)} = V_t^{(TD)}$.*

*Proof.* We use induction to prove this statement.

**Base case:** When $t = 0$, the PT-estimate of state $s$ is $V_0^{(PT)}(s) = V^{(P)}(s) + V_0^{(T)}(s) = V^{(P)}(s)$. And, the TD estimate of state $s$ is $V_0^{(TD)}(s) = V^{(P)}(s)$. Therefore, both PT and TD estimates are equal.

**Induction hypothesis:** PT and TD estimates are equal for some timestep $k$ for all states. That is, $V_k^{(PT)}(s) = V^{(P)}(s) + V_k^{(T)}(s) = V_k^{(TD)}(s)$.

**Induction step:** PT estimates at timestep $k+1$ is

$$V_{k+1}^{(PT)}(s) = V^{(P)}(s) + V_{k+1}^{(T)}(s),$$
$$= V^{(P)}(s) + V_k^{(T)}(s) + \alpha_{k+1}(R_{k+1} + \gamma(V^{(P)}(S_{k+1}) + V_t^{(T)}(S_{k+1})) - V^{(P)}(s) - V_k^{(T)}(s)),$$

$$= V_k^{(PT)}(s) + \alpha_{k+1}(R_{k+1} + \gamma V_k^{(PT)}(S_{k+1}) - V_k^{(PT)}(s)),$$
$$= V_k^{(TD)}(s) + \alpha_{k+1}(R_{t+1} + \gamma V_k^{(TD)}(S_{t+1}) - V_k^{(TD)}(s)),$$
$$\boxed{= V_{k+1}^{(TD)}(s).}$$

The penultimate step follows from the induction hypothesis completing the proof. $\qquad\square$

**Theorem B.1.2.** *The expected target in the transient value function update rule is a contraction-mapping Bellman operator:* $\mathcal{T}^{(T)}V^{(T)} = r_\pi + \gamma \mathcal{P}_\pi V^{(P)} - V^{(P)} + \gamma \mathcal{P}_\pi V^{(T)}.$

*Proof.* The target for state $s$ at timestep $t$ in the transient value function update rule is $R_{t+1} + \gamma V^{(P)}(S_{t+1}) - V^{(P)}(s) + \gamma V^{(T)}(S_{t+1})$. The expected target is:

$$\mathcal{T}^{(T)}V^{(T)}(s) = \mathbb{E}_\pi[R_{t+1} + \gamma V^{(P)}(S_{t+1}) - V^{(P)}(s) + \gamma V^{(T)}(S_{t+1})|S_t = s],$$
$$= r_\pi(s) + \gamma(\mathcal{P}_\pi V^{(P)})(s) - V^{(P)}(s) + \gamma(\mathcal{P}_\pi V^{(T)})(s).$$

We get the operator by expressing the above equation in the vector form:

$$\mathcal{T}^{(T)}V^{(T)} = r_\pi + \gamma \mathcal{P}_\pi V^{(P)} - V^{(P)} + \gamma \mathcal{P}_\pi V^{(T)}.$$

For contraction proof, consider two vectors $u, v \in \mathbb{R}^{|\mathcal{S}|}$.

$$\left\| \mathcal{T}^{(T)}u - \mathcal{T}^{(T)}v \right\|_\infty = \left\| r_\pi + \gamma \mathcal{P}_\pi V^{(P)} - V^{(P)} + \gamma \mathcal{P}_\pi u - (r_\pi + \gamma \mathcal{P}_\pi V^{(P)} - V^{(P)} + \gamma \mathcal{P}_\pi v) \right\|_\infty,$$
$$= \left\| \gamma \mathcal{P}_\pi(u - v) \right\|_\infty,$$
$$\boxed{\leq \gamma \left\| u - v \right\|_\infty.}$$

Therefore, the transient value function Bellman operator is a contraction mapping with contraction factor $\gamma$. $\qquad\square$

**Theorem B.1.3.** *The unique fixed point of the transient operator is* $(I - \gamma \mathcal{P}_\pi)^{-1}r_\pi - V^{(P)}$.

*Proof.* We get the fixed point, $V_\pi^{(T)}$, by repeatedly applying the operator to a vector $v$:

$$V_\pi^{(T)} = r_\pi + \gamma \mathcal{P}_\pi V^{(P)} - V^{(P)} + \gamma \mathcal{P}_\pi v,$$
$$= r_\pi + \gamma \mathcal{P}_\pi V^{(P)} - V^{(P)} + \gamma \mathcal{P}_\pi(r_\pi + \gamma \mathcal{P}_\pi V^{(P)} - V^{(P)} + \gamma \mathcal{P}_\pi v),$$
$$= r_\pi + \gamma \mathcal{P}_\pi r_\pi + (\gamma \mathcal{P}_\pi)^2 V^{(P)} - V^{(P)} + (\gamma \mathcal{P}_\pi)^2 v,$$
$$= r_\pi + \gamma \mathcal{P}_\pi r_\pi + (\gamma \mathcal{P}_\pi)^2 V^{(P)} + \cdots + (\gamma \mathcal{P}_\pi)^n V^{(P)} - V^{(P)} + (\gamma \mathcal{P}_\pi)^n v,$$
$$\boxed{= (I - \gamma \mathcal{P}_\pi)^{-1}r_\pi - V^{(P)}.}$$

We applied the operator to $v$ in steps 1 to 4 and used the property $\lim_{n\to\infty}(\gamma \mathcal{P}_\pi)^n \to 0$ in the penultimate step as $\gamma \mathcal{P}_\pi$ is a sub-stochastic matrix. Many terms cancel out as it is a telescoping sum. $\qquad\square$

**Theorem B.1.4.** *Let* $\mathcal{M}$ *be an MDP in which the agent is performing transient value function updates for* $\pi$. *Define an MRP,* $\widetilde{\mathcal{M}}$, *with state space* $\widetilde{\mathcal{S}} : \{(S_t, A_t, S_{t+1}), \forall t\}$, *transition dynamics* $\widetilde{\mathcal{P}}(x'|x, a') = \mathcal{P}(s''|s', a'), \forall x, x' \in \widetilde{\mathcal{S}}, \forall s', s'' \in \mathcal{S}, \forall a' \in \mathcal{A}$ *and reward function* $\widetilde{\mathcal{R}}(x, a') = \mathcal{R}(s, a) + \gamma V^{(P)}(s') - V^{(P)}(s)$. *Then, the fixed point of Eq.(5) is the value function of* $\pi$ *in* $\widetilde{\mathcal{M}}$.

*Proof.* The state space of the new MDP $\widetilde{\mathcal{M}}$ is made up of all possible consecutive tuples of $(S_t, A_t, S_{t+1})$ pairs. The rewards in this MDP are defined as $\widetilde{\mathcal{R}}(x, a') = \mathcal{R}(s, a) + \gamma V^{(P)}(s') - V^{(P)}(s), \forall x \in \widetilde{\mathcal{S}}, \forall a' \in \mathcal{A}$. The transition matrix $\widetilde{\mathcal{P}}(x'|x, a') = prob(x'|x, a') = prob(s', a', s''|s, a, s', a') = prob(s''|s', a') = \mathcal{P}(s''|s', a')$ using Markov property of $\mathcal{M}$. Then, the value of state $x = (s, a, s')$ is:

$$v_\pi(x) = v_\pi(s, a, s') = \mathbb{E}_\pi[Z_t|X_t = (s, a, s')],$$

where $Z_t = \sum_{k=t}^{\infty} \gamma^{k-t}\tilde{R}_{k+1}$ is the returns starting from state $X_t = (s, a, s')$. Without loss of generality, consider a sample trajectory generated according to $\pi$, $\tau_k = (S_0 = s, A_0 = a, R_1, S_1 = s', A_1, R_2, S_2, A_2, R_3, \ldots, R_T, S_T)$ where $S_T$ is the terminal state. The returns $Z^{\tau_k}$ of this trajectory is:

$$Z^{\tau_k} = \sum_{k=0}^{\infty} \gamma^k \tilde{R}_{k+1}$$

$$= \sum_{k=0}^{\infty} \gamma^k (R_{k+1} + \gamma V^{(P)}(S_{k+1}) - V^{(P)}(S_k)),$$

$$= \sum_{k=0}^{\infty} \gamma^k R_{k+1} - V^{(P)}(S_0 = s),$$

$$\boxed{= G^{\tau_k} - V^{(P)}(s).}$$

Returns of every trajectory end up being the difference between the returns in the original MDP, $G^{\tau_k}$ and the permanent value function. Most importantly, these returns only depend on the state $s$ and it is the same for all $a$ and $s'$ choices following $s$. The value of state $x = (s, a, s')$ can then be simplified using this observation:

$$v_\pi(s, a, s') = \mathbb{E}_\tau[Z_t | X_t = (s, a, s')],$$

$$= \mathbb{E}_\tau[G_t - V^{(P)}(s) | S_t = s, A_t = \cdot, S_{t+1} = \cdot],$$

$$\boxed{= V_\pi^{(T)}(s).}$$

$\square$

## B.2 Permanent Value Function Results

We focus on permanent value function in the next two theorems. First, we characterize the fixed point of the permanent value function. We use assumption 1 on the task distribution for the following proofs:

**Theorem B.2.1.** *Following Theorem 2, under assumption 1 and Robbins-Monro step-size conditions, the sequence of updates computed by Eq.(4) contracts to a unique fixed point $\mathbb{E}_\tau[v_\tau]$.*

*Proof.* The permanent value function is updated using Eq. (4) and samples stored in the buffer $\mathcal{B}$ at the end of each task. This update rule can be simplified as:

$$V_{k+1}^{(P)}(s) \leftarrow V_k^{(P)}(s) + \overline{\alpha}_k \sum_{i=1}^{L} \mathbb{I}_{(s_i=s)}(V_\tau^{(PT)}(s) - V_k^{(P)}(s)),$$

$$= V_k^{(P)}(s) + \overline{\alpha}_k(V_\tau^{(PT)}(s) - V_k^{(P)}(s))\sum_{i=1}^{L}\mathbb{I}_{(s_i=s)},$$

$$= V_k^{(P)}(s) + \overline{\alpha}_k L(s)(V_\tau^{(PT)}(s) - V_k^{(P)}(s)),$$

$$= V_k^{(P)}(s) + \widetilde{\alpha}_k(s)(V_\tau^{(PT)}(s) - V_k^{(P)}(s)),$$

where $L$ is the length of the buffer, $\mathbb{I}$ is the indicator function, $L(s)$ is the number of samples for state $s$ in the buffer $\mathcal{B}$, $\widetilde{\alpha}_k(s) = \overline{\alpha}_k L(s)$. The above update is performed once for every state in $\mathcal{B}$ at the end of each task. We assume that the buffer has enough samples for each state so that all states are updated. This can be ensured by storing all the samples in $\mathcal{B}$ for a long but finite period of time. The updates performed to permanent value function until that timestep will not affect the fixed point since we are in the tabular setting.

Let the sequence of learning rate $\widetilde{\alpha}_k(s)$ obey the conditions: $\sum_{n=1}^{\infty} \widetilde{\alpha}_n(s) = \infty$ and $\sum_{n=1}^{\infty} \widetilde{\alpha^2}_n(s) = 0$.

We can rewrite the PM update rule by separating the noise term:

$$V_{k+1}^{(P)}(s) \leftarrow V_k^{(P)}(s) + \widetilde{\alpha}_k(s)(V_\tau^{(PT)}(s) - V_k^{(P)}(s)),$$

$$= V_k^{(P)}(s) + \widetilde{\alpha}_k(s)(\mathbb{E}[V_\tau^{(PT)}(s)] - V_k^{(P)}(s) + V_\tau^{(PT)}(s) - \mathbb{E}[V_\tau^{(PT)}(s)]),$$
$$= V_k^{(P)}(s) + \widetilde{\alpha}_k(s)(\mathbb{E}[V_\tau^{(PT)}(s)] - V_k^{(P)}(s) + w_k).$$

Let $\mathcal{F}_k$ denote the history of the algorithm. Then, $\mathbb{E}[w_k(s)|\mathcal{F}_k] = \mathbb{E}[V_\tau^{(PT)}(s) - \mathbb{E}[V_\tau^{(PT)}(s)]] = 0$ and under the assumption $\mathbb{E}[w_k^2(s)|\mathcal{F}_k] \leq A + B V_k^{(P)^2}(s)$ (finite noise variance), we conclude that the permanent value function converges to $E[V_\tau^{(PT)}(s)]$. But $E[V_\tau^{(PT)}(s)]$ is contracting towards $\mathbb{E}[v_\tau(s)]$, therefore, the permanent value function is contracting towards $\mathbb{E}[v_\tau(s)]$. $\qquad\square$

**Theorem B.2.2.** *The fixed point of the permanent value function optimizes the jumpstart objective,* $J = \arg\min_{u \in \mathbb{R}^{|S|}} \frac{1}{2}\mathbb{E}_\tau[\|u - v_\tau\|_2^2].$

*Proof.* The jumpstart objective function is given by $J = \arg\min_{u \in \mathbb{R}^{|S|}} \frac{1}{2}\mathbb{E}_\tau[\|u - v_\tau\|_2^2]$. Let $K = \frac{1}{2}\mathbb{E}_\tau[\|u - v_\tau\|_2^2]$. We get the optimal point by differentiating $K$ with respect to $u$ and then equating it to 0:

$$\nabla_u K = \nabla_u \left( \frac{1}{2}\mathbb{E}_\tau[\|u - v_\tau\|_2^2] \right),$$
$$= \frac{1}{2}\mathbb{E}_\tau[\nabla_u \|u - v_\tau\|_2^2],$$
$$= \frac{1}{2}\mathbb{E}_\tau[2(u - v_\tau)],$$
$$= u - \mathbb{E}_\tau[v_\tau],$$
$$0 = u - \mathbb{E}_\tau[v_\tau],$$
$$\boxed{u = \mathbb{E}_\tau[v_\tau].}$$

Therefore, the fixed point of the permanent value function also optimizes the jumpstart objective. $\qquad\square$

## B.3 Semi-Continual RL Results

We assume that at time $t$, both $V^{(TD)}$ and $V^{(PT)}$ have converged to the true value function of task $i$ and the agent gets samples from task $\tau$ from timestep $t+1$ onward for the following theorems.

**Theorem B.3.1.** *Under assumption 1, there exists $k_0$ such that $\forall k \geq k_0$, $\mathbb{E}_\tau[\left\|V_{t+k}^{(TD)} - v_i\right\|_2^2] > \mathbb{E}_\tau[\left\|V^{(P)} - v_i\right\|_2^2], \forall i.$*

*Proof.* Let $\left\|V^{(P)} - v_i\right\|_2^2 = D_i$, which is a constant because we are updating transient value function only. Let $\|v_\tau - v_i\|_2^2 = \Delta_{\tau,i}$ denote the distance between value functions of two tasks $i$ and $\tau$. Clearly, $\Delta_{\tau,i} = \Delta_{i,\tau} \neq 0$ and $\Delta_{i,i} = 0$. And, let $Err_t^{i,\tau} = \left\|V_t^{(TD)} - v_i\right\|_2^2$ denote prediction error with respect to the previous task $i$ at timestep $t$ for TD estimates.

At timestep $t$, $V^{(PT)} = V^{(TD)} = v_i$. The agent starts getting samples from task $\tau$ from timestep $t+1$. Using the result from Bhandari et al. [6]:

$$\left\|V_{t+1}^{(TD)} - v_\tau\right\|_2^2 \leq \exp\left\{ \frac{-\omega(1-\gamma^2)}{4} \right\} \left\|V_t^{(TD)} - v_\tau\right\|_2^2,$$
$$= \exp\left\{ \frac{-(1-\gamma^2)}{4} \right\} \Delta_{\tau,i},$$

since the smallest eigenvalue of the feature matrix $\omega = 1$ for tabular approximations. Let $C = \exp\{\frac{-(1-\gamma^2)}{4}\}$. Now we can apply the above result recursively for $k$ timesteps to get

$$\left\|V_{t+k}^{(TD)} - v_\tau\right\|_2^2 \leq C^k \Delta_{\tau,i}.$$

Then the error with respect to the past task after $k$ timesteps is,

$$Err_{t+k}^{i,\tau} \geq \Delta_{\tau,i} - C^k \Delta_{\tau,i} = \Delta_{\tau,i}(1 - C^k).$$

Taking the expectation with respect to $\tau$ gives

$$\mathbb{E}_\tau[Err_{t+k}^{i,\tau}] = \sum_\tau p_\tau Err_{t+k}^{i,\tau},$$
$$\geq \sum_\tau p_\tau \Delta_{\tau,i}(1 - C^k),$$
$$\geq (1 - C^k)\mathbb{E}_\tau[\Delta_{\tau,i}].$$

Now we will find a timestep $k_0$ such that $\forall k > k_0$, $E_\tau[Err_{t+k}^{i,\tau}] > D_i$. Let $k_i$ denote the timestep for which $(1 - C^{k_i})E_\tau[\Delta_{\tau,i}] = D_i$. Then,

$$(1 - C^{k_i})E_\tau[\Delta_{\tau,i}] = D_i,$$
$$(1 - C^{k_i}) = \frac{D_i}{E_\tau[\Delta_{\tau,i}]},$$
$$C^{k_i} = \frac{E_\tau[\Delta_{\tau,i}] - D_i}{E_\tau[\Delta_{\tau,i}]},$$

$$\boxed{k_i = \frac{1}{\log C} \log\left(\frac{E_\tau[\Delta_{\tau,i}] - D_i}{E_\tau[\Delta_{\tau,i}]}\right).}$$

So, after time $t + k_i$, $V^{(P)}$ is a better estimate of the value function of the past task $i$ than the current TD estimates. Then, by letting $k_0 = \max_i k_i$ we have $\mathbb{E}_\tau[Err_{t+k_0}^{i,\tau}] > D_i, \forall i$.

Note that the above result depends on the fact that $k_i$ exists, which is only possible when $E_\tau[\Delta_{\tau,i}] - D_i > 0$. We will now show that there exists at least one solution, the fixed point of the permanent value function ($\mathbb{E}_\tau[v_\tau]$), which satisfies this constraint.

$$E_\tau[\Delta_{\tau,i}] > \|\mathbb{E}_\tau[v_i - v_\tau]\|_2^2, \text{(Using Jensen's inequality)}$$
$$= \|v_i - \mathbb{E}_\tau[v_\tau]\|_2^2.$$

Then,

$$\boxed{E_\tau[\Delta_{\tau,i}] - D_i > \|v_i - \mathbb{E}_\tau[v_\tau]\|_2^2 - D_i = 0.}$$

The final step follows from the fact that $D_i = \left\|V^{(P)} - v_i\right\|_2^2 = \|\mathbb{E}_\tau[v_\tau] - v_i\|_2^2$ for $V^{(P)}$ fixed point. $\qquad\square$

Next, we show that our method has tighter upper bound on errors compared with the TD learning algorithm in expectation.

**Theorem B.3.2.** *Under assumption 1, there exists $k_0$ such that $\forall k \leq k_0$, $\mathbb{E}_\tau\left[\left\|v_\tau - V_{t+k}^{(PT)}\right\|_2^2\right]$ has a tighter upper bound compared with $\mathbb{E}_\tau\left[\left\|v_\tau - V_{t+k}^{(TD)}\right\|_2^2\right], \forall i.$*

*Proof.* We assume that $V^{(P)}$ is converged to its fixed point, $\mathbb{E}_\tau[v_\tau]$ (Theorem 5). Using sample complexity bounds from Bhandari et al. [6] for TD learning algorithm, we get:

$$\mathbb{E}_\tau\left[\left\|v_\tau - V_{t+k}^{(TD)}\right\|_2^2\right] \leq \mathbb{E}_\tau\left[\exp\left\{-\left(\frac{(1-\gamma)^2\omega}{4}\right)T\right\}\left\|v_\tau - V_t^{(TD)}\right\|_2^2\right],$$
$$= \exp\left\{-\left(\frac{(1-\gamma)^2\omega}{4}\right)T\right\}\mathbb{E}_\tau\left[\left\|v_\tau - V_t^{(TD)}\right\|_2^2\right].$$

We know that our algorithm can be interpreted as TD learning with $V^{(P)}$ as the starting point (see Theorem 1). Using this relationship, we can apply the sample complexity bound to get:

$$\mathbb{E}_\tau \left[ \left\| v_\tau - V_{t+k}^{(PT)} \right\|_2^2 \right] \leq \mathbb{E}_\tau \left[ \exp \left\{ -\left( \frac{(1-\gamma)^2 \omega}{4} \right) T \right\} \left\| v_\tau - V_t^{(P)} \right\|_2^2 \right],$$

$$= \exp \left\{ -\left( \frac{(1-\gamma)^2 \omega}{4} \right) T \right\} \mathbb{E}_\tau \left[ \left\| v_\tau - V_t^{(P)} \right\|_2^2 \right].$$

For $k = 0$, we know that our algorithm has lower error in expectation (see Theorem 6), that is $\mathbb{E}_\tau \left[ \left\| v_\tau - V_t^{(P)} \right\|_2^2 \right] \leq \mathbb{E}_\tau \left[ \left\| v_\tau - V_t^{(TD)} \right\|_2^2 \right]$. Therefore, the error bound for our algorithm is tighter compared with that of TD learning algorithm. As $k \to \infty$, the right hand side of both the equations goes to 0, which confirms that both algorithms converges to the true value function. $\qquad \square$

The theorem shows that the estimates for our algorithm lie within a smaller region compared to that of TD learning estimates. This doesn't guarantee low errors for our algorithms, but we can expect it to be closer to the true value function.

The next theorems contain analytical MSE equations for TD-learning and for our algorithm. Our analysis is similar to that of [42], but we don't assume that rewards are only observed in the final transition. Analytical MSE equations allow calculating the exact MSE that an algorithm would incur at each timestep when applied to a problem, instead of averaging the performance over several seeds, which is the first step towards other analyses such as bias-variance decomposition of value estimates, error bounds and bias-variance bounds, step-size adaptation for the update rule, etc. We use these equations to compare our method against TD-learning when varying the task distribution diameter (Sec. C.2).

We assume that the agent gets $N$ i.i.d samples $\langle s, r, s' \rangle$ from task $\tau$ in each round, where $s$ is sampled according to the stationary distribution $d_\tau$. Also, the agent accumulates errors from all the samples and updates its estimates at the end of the round $t$. Let $\mathbb{E}_{seeds}$ denote the expectation with respect to random seeds, i.e., expectation with respect to the data generation process which generates the data that the agent observes over rounds and within a round.

**Theorem B.3.3.** *Let:*

$$m_t^{(TD)}(s) = \mathbb{E}_{seeds}[V_t^{(TD)}(s)],$$

$$Cov(V_t^{(TD)}(s), V_t^{(TD)}(x)) = \Sigma_t^{(TD)}(s, x) - m_t^{(TD)}(s) m_t^{(TD)}(x),$$

*where $\Sigma_t^{(TD)}(s, x) = \mathbb{E}_{seeds}[V_t^{(TD)}(s) V_t^{(TD)}(s)]$. We have:*

$$m_{t+1}^{(TD)}(s) = m_t^{(TD)}(s) + \alpha N d_\pi(s) \Delta_t^{(TD)}(s),$$

$$\Sigma_{t+1}^{(TD)}(s, x) = \Sigma_t^{(TD)}(s, x) + \alpha \Omega_t^{(TD)}(s, x) + \alpha^2 \Psi_t^{(TD)}(s, x),$$

*and, the mean squared error $\xi_t^{(TD)}$ at timestep $t$ is*

$$\xi_{t+1}^{(TD)} = \xi_t^{(TD)} + \alpha \sum_{s \in \mathcal{S}} d_\pi(s)(-2 v_\pi(s) \Delta_t^{(TD)}(s) + \Omega_t^{(TD)}(s, s))$$

$$+ \alpha^2 \sum_{s \in \mathcal{S}} d_\pi(s) \Psi_t^{(TD)}(s, s).$$

*Proof.*

$$m_{t+1}^{(TD)}(s) = \mathbb{E}_{seeds}[V_{t+1}^{(TD)}(s)],$$

$$= \mathbb{E}_{seeds} \left[ V_t^{(TD)}(s) + \alpha \sum_{k=1}^N \mathbb{I}_{S_k=s}(R_k + \gamma V_t^{(TD)}(S_k') - V_t^{(TD)}(s)) \right],$$

$$= m_t^{(TD)}(s) + \alpha \sum_{k=1}^N \mathbb{E}_{seeds}[\mathbb{I}_{S_k=s}(R_k + \gamma V_t^{(TD)}(S_k') - V_t^{(TD)}(s))],$$

$$= m_t^{(TD)}(s) + \alpha \sum_{k=1}^{N} \mathbb{E}_{seeds} \left[ \mathbb{E}[\mathbb{I}_{S_k=s}(R_k + \gamma V_t^{(TD)}(S_k') - V_t^{(TD)}(s))|seed] \right],$$

$$= m_t^{(TD)}(s) + \alpha \sum_{k=1}^{N} \mathbb{E}_{seeds} \left[ \widehat{d_\pi}(s) \mathbb{E}[(R_k + \gamma V_t^{(TD)}(S_k') - V_t^{(TD)}(s))|seed, S_k = s] \right],$$

$$= m_t^{(TD)}(s) + \alpha \sum_{k=1}^{N} \mathbb{E}_{seeds} \left[ \underbrace{\widehat{d_\pi}(s)(\widehat{r_\pi}(s) + \gamma(\widehat{\mathcal{P}_\pi = V_t^{(TD)}})(s) - V_t^{(TD)}(s))}_{\text{Independent quantities}} \right],$$

$$= m_t^{(TD)}(s) + \alpha \sum_{k=1}^{N} d_\pi(s)(r_\pi(s) + \gamma(\mathcal{P}_\pi m_t^{(TD)})(s) - m_t^{(TD)}(s))],$$

$$= m_t^{(TD)}(s) + \alpha N d_\pi(s)(r_\pi(s) + \gamma(\mathcal{P}_\pi m_t^{(TD)})(s) - m_t^{(TD)}(s))],$$

$$\boxed{m_{t+1}^{(TD)}(s) = m_t^{(TD)}(s) + \alpha \Delta_t^{(TD)}(s),}$$

where $\Delta_t^{(TD)}(s) = r_\pi(s) + \gamma(\mathcal{P}_\pi m_t^{(TD)})(s) - m_t^{(TD)}(s)$. We used the following results in the proof:

1. $\mathbb{E}_{seeds}[\widehat{d_\pi}(s)\widehat{r_\pi}(s)] = \mathbb{E}_{seeds}[\widehat{d_\pi}(s)] \mathbb{E}_{seeds}[\widehat{r_\pi}(s)] = d_\pi(s)r_\pi(s).$

2. $\mathbb{E}_{seeds}[\widehat{d_\pi}(s)\gamma(\widehat{\mathcal{P}_\pi V_t^{(TD)}})(s)] = \gamma \mathbb{E}_{seeds}[\widehat{d_\pi}(s)] \mathbb{E}_{seeds}[(\widehat{\mathcal{P}_\pi V_t^{(TD)}})(s)],$

$$= \gamma d_\pi(s) \sum_{s'} \mathbb{E}_{seeds}[\widehat{\mathcal{P}_\pi}(s'|s)V_t^{(TD)}(s')],$$

$$= \gamma d_\pi(s) \sum_{s'} \mathcal{P}_\pi(s'|s)m_t^{(TD)}(s'),$$

$$= \gamma d_\pi(s)(\mathcal{P}_\pi m_t^{(TD)})(s).$$

3. $\mathbb{E}_{seeds}[\widehat{d_\pi}(s)V_t^{(TD)}(s)] = \mathbb{E}_{seeds}[\widehat{d_\pi}(s)] \mathbb{E}_{seeds}[V_t^{(TD)}(s)] = d_\pi(s)m_t^{(TD)}(s).$

Now consider $\Sigma_{t+1}^{(TD)}(s, x)$,

$$\Sigma_{t+1}^{(TD)}(s, x) = \mathbb{E}_{seeds}[V_{t+1}^{(TD)}(s)V_{t+1}^{(TD)}(x)],$$

$$= \mathbb{E}_{seeds} \left[ \left( V_t^{(TD)}(s) + \alpha \sum_{k=1}^{N} \mathbb{I}_{S_k=s}(R_k + \gamma V_t^{(TD)}(S_k') - V_t^{(TD)}(s))V_{t+1}^{(TD)}(x) \right) \right.$$

$$\left. \left( V_t^{(TD)}(x) + \alpha \sum_{l=1}^{N} \mathbb{I}_{S_l=x}(R_l + \gamma V_t^{(TD)}(X_l') - V_t^{(TD)}(x)) \right) \right],$$

$$= \mathbb{E}_{seeds}[V_t^{(TD)}(s)V_t^{(TD)}(x)]$$

$$+ \alpha \mathbb{E}_{seeds} \left[ V_t^{(TD)}(s) \left( \sum_{l=1}^{N} \mathbb{I}_{S_l=x}(R_l + \gamma V_t^{(TD)}(X_l') - V_t^{(TD)}(x)) \right) \right] \qquad (9)$$

$$+ \alpha \mathbb{E}_{seeds} \left[ V_t^{(TD)}(x) \left( \sum_{k=1}^{N} \mathbb{I}_{S_k=s}(R_k + \gamma V_t^{(TD)}(S_k') - V_t^{(TD)}(s)) \right) \right] \qquad (10)$$

$$+ \alpha^2 \mathbb{E}_{seeds} \left[ \left( \sum_{k=1}^{N} \mathbb{I}_{S_k=s}(R_k + \gamma V_t^{(TD)}(S_k') - V_t^{(TD)}(s)) \right) \right.$$

$$\left. \left( \sum_{l=1}^{N} \mathbb{I}_{S_l=x}(R_l + \gamma V_t^{(TD)}(X_l') - V_t^{(TD)}(x)) \right) \right] \qquad (11)$$

Next, we will tackle each part separately. We will begin with Eq (9):

$$\mathbb{E}_{seeds}\left[V_t^{(TD)}(s)\left(\sum_{l=1}^N \mathbb{I}_{S_l=x}(R_l + \gamma V_t^{(TD)}(X_l') - V_t^{(TD)}(x))\right)\right]$$

$$= \mathbb{E}_{seeds}\left[V_t^{(TD)}(s)\sum_{l=1}^N \mathbb{E}[(\mathbb{I}_{S_l=x}(R_l + \gamma V_t^{(TD)}(X_l') - V_t^{(TD)}(x)))|seed]\right],$$

$$= \mathbb{E}_{seeds}[V_t^{(TD)}(s)N\widehat{d_\pi}(x)(\widehat{r_\pi}(x) + \gamma(\widehat{\mathcal{P}_\pi}V_t^{(TD)})(x) - V^{(TD)}(x))],$$

$$= N\mathbb{E}_{seeds}[\widehat{d_\pi}(x)\widehat{r_\pi}(x)V_t^{(TD)}(s)] + \gamma N\mathbb{E}_{seeds}[\widehat{d_\pi}(x)(\widehat{\mathcal{P}_\pi}V_t^{(TD)})(x)V_t^{(TD)}(s)]$$
$$- N\mathbb{E}_{seeds}[\widehat{d_\pi}(x)V_t^{(TD)}(x)V_t^{(TD)}(s)],$$

$$= Nd_\pi(x)r_\pi(x)m^{(TD)}(s) + \gamma Nd_\pi(x)\mathbb{E}_{seeds}\left[\sum_{x'}\widehat{\mathcal{P}_\pi}(x'|x)V_t^{(TD)}(x')V_t^{(TD)}(x)\right]$$

$$- Nd_\pi(x)\Sigma_t^{(TD)}(s,x),$$

$$= Nd_\pi(x)\left(r_\pi(x)m_t^{(TD)}(s) + \gamma\sum_{x'}\mathbb{E}_{seeds}[\widehat{\mathcal{P}_\pi}(x'|x)V_t^{(TD)}(x')V_t^{(TD)}(x)] - \Sigma_t^{(TD)}(s,x)\right),$$

$$= Nd_\pi(x)\left(r_\pi(x)m_t^{(TD)}(s) + \gamma\sum_{x'}\mathcal{P}_\pi(x'|x)\Sigma_t^{(TD)}(x',s) - \Sigma_t^{(TD)}(s,x)\right),$$

$$\boxed{= Nd_\pi(x)\left(r_\pi(x)m_t^{(TD)}(s) + \gamma(\mathcal{P}_\pi\Sigma_t^{(TD)})(x,s) - \Sigma_t^{(TD)}(s,x)\right).}$$

A similar simplification can be done to Eq. (10) which results in:

$$\boxed{(10) = Nd_\pi(s)\left(r_\pi(s)m^{(TD)}(x) + \gamma(\mathcal{P}_\pi\Sigma_t^{(TD)})(s,x) - \Sigma_t^{(TD)}(x,s)\right).}$$

We will break down Eq. (11) into 9 parts and simplify each of them separately.

$$(11) = \mathbb{E}_{seeds}\left[\left(\sum_{k=1}^N \mathbb{I}_{S_k=s}R_k\right)\left(\sum_{l=1}^N \mathbb{I}_{S_l=x}R_l\right)\right] \tag{12}$$

$$+ \gamma\mathbb{E}_{seeds}\left[\left(\sum_{k=1}^N \mathbb{I}_{S_k=s}R_k\right)\left(\sum_{l=1}^N \mathbb{I}_{S_l=x}V_t^{(TD)}(X_l')\right)\right] \tag{13}$$

$$- \mathbb{E}_{seeds}\left[\left(\sum_{k=1}^N \mathbb{I}_{S_k=s}R_k\right)\left(\sum_{l=1}^N \mathbb{I}_{S_l=x}V_t^{(TD)}(x)\right)\right] \tag{14}$$

$$+ \gamma\mathbb{E}_{seeds}\left[\left(\sum_{k=1}^N \mathbb{I}_{S_k=s}V_t^{(TD)}(S_k')\right)\left(\sum_{l=1}^N \mathbb{I}_{S_l=x}R_l\right)\right] \tag{15}$$

$$+ \gamma^2\mathbb{E}_{seeds}\left[\left(\sum_{k=1}^N \mathbb{I}_{S_k=s}V_t^{(TD)}(S_k')\right)\left(\sum_{l=1}^N \mathbb{I}_{S_l=x}V_t^{(TD)}(X_l')\right)\right] \tag{16}$$

$$- \gamma\mathbb{E}_{seeds}\left[\left(\sum_{k=1}^N \mathbb{I}_{S_k=s}V_t^{(TD)}(S_k')\right)\left(\sum_{l=1}^N \mathbb{I}_{S_l=x}V_t^{(TD)}(x)\right)\right] \tag{17}$$

$$- \mathbb{E}_{seeds}\left[\left(\sum_{k=1}^N \mathbb{I}_{S_k=s}V_t^{(TD)}(s)\right)\left(\sum_{l=1}^N \mathbb{I}_{S_l=x}R_l\right)\right] \tag{18}$$

$$- \gamma\mathbb{E}_{seeds}\left[\left(\sum_{k=1}^N \mathbb{I}_{S_k=s}V_t^{(TD)}(s)\right)\left(\sum_{l=1}^N \mathbb{I}_{S_l=x}V_t^{(TD)}(X_l')\right)\right] \tag{19}$$

$$+ \mathbb{E}_{seeds}\left[\left(\sum_{k=1}^N \mathbb{I}_{S_k=s}V_t^{(TD)}(s)\right)\left(\sum_{l=1}^N \mathbb{I}_{S_l=x}V_t^{(TD)}(x)\right)\right]. \tag{20}$$

Consider Eq. (12)

$$\mathbb{E}_{seeds}\left[\left(\sum_{k=1}^{N}\mathbb{I}_{S_k=s}R_k\right)\left(\sum_{l=1}^{N}\mathbb{I}_{S_l=x}R_l\right)\right] = \mathbb{E}_{seeds}\left[\mathbb{E}\left[\left(\sum_{k=1}^{N}\mathbb{I}_{S_k=s}R_k\right)\left(\sum_{l=1}^{N}\mathbb{I}_{S_l=x}R_l\right)|seed\right]\right],$$

$$= \mathbb{E}_{seeds}\left[\mathbb{E}\left[\left(\sum_{k=1}^{N}\mathbb{I}_{S_k=s}R_k\right)\left(\sum_{\substack{l=1,\\l\neq k}}^{N}\mathbb{I}_{S_l=x}R_l\right)|seed\right]\right] + \mathbb{E}_{seeds}\left[\mathbb{E}\left[\left(\sum_{m=1}^{N}\mathbb{I}_{S_m=s}\mathbb{I}_{S_m=x}R_mR_m\right)|seed\right]\right],$$

$$= \mathbb{E}_{seeds}\left[\mathbb{E}\left[\left(\sum_{k=1}^{N}\mathbb{I}_{S_k=s}R_k\right)|seed\right]\mathbb{E}\left[\left(\sum_{\substack{l=1,\\l\neq k}}^{N}\mathbb{I}_{S_l=x}R_l\right)|seed\right]\right] + \mathbb{I}_{s=x}\mathbb{E}_{seeds}\left[\sum_{m=1}^{N}\widehat{d_\pi}(s)\mathbb{E}[R_m^2|seed,S_m=s]\right],$$

$$= \mathbb{E}_{seeds}\left[\sum_{k=1}^{N}\widehat{d_\pi}(s)\widehat{r_\pi}(s)\sum_{\substack{l=1,\\l\neq k}}^{N}\widehat{d_\pi}(x)\widehat{r_\pi}(x)\right] + \mathbb{I}_{s=x}\sum_{m=1}^{N}\mathbb{E}_{seeds}[\widehat{d_\pi}(s)(\widehat{c(s)} + \widehat{r_\pi}^2(s))],$$

$$= \sum_{k=1}^{N}\sum_{\substack{l=1,\\l\neq k}}^{N}\mathbb{E}_{seeds}[\widehat{d_\pi}(s)\widehat{r_\pi}(s)\widehat{d_\pi}(x)\widehat{r_\pi}(x)] + \mathbb{I}_{s=x}Nd_\pi(s)\left(c(s) + r_\pi^2(s)\right),$$

$$\boxed{= N(N-1)d_\pi(s)d_\pi(x)r_\pi(s)r_\pi(x) + \mathbb{I}_{s=x}Nd_\pi(s)\left(c(s) + r_\pi^2(s)\right).}$$

We have used the i.i.d assumption to separate the terms. Later, we used the indicator function to indicate the part which is non-zero. Also, we have used the definition of the second moment in the penultimate step.

We do a similar simplification to Eq. (13) now.

$$\gamma\mathbb{E}_{seeds}\left[\left(\sum_{k=1}^{N}\mathbb{I}_{S_k=s}R_k\right)\left(\sum_{l=1}^{N}\mathbb{I}_{S_l=x}V_t^{(TD)}(X_l')\right)\right]$$

$$= \gamma\mathbb{E}_{seeds}\left[\sum_{k=1}^{N}\sum_{\substack{l=1,\\l\neq k}}^{N}\mathbb{I}_{S_k=s}\mathbb{I}_{S_l=x}R_kV_t^{(TD)}(X_l')\right] + \gamma\mathbb{E}_{seeds}\left[\sum_{m=1}^{N}\mathbb{I}_{S_m=s}\mathbb{I}_{S_m=x}R_mV_t^{(TD)}(X_m')\right],$$

$$= \gamma\mathbb{E}_{seeds}\left[\sum_{k=1}^{N}\sum_{\substack{l=1,\\l\neq k}}^{N}\mathbb{E}[\mathbb{I}_{S_k=s}\mathbb{I}_{S_l=x}R_kV_t^{(TD)}(X_l')|seed]\right] + \gamma\mathbb{I}_{s=x}\sum_{m=1}^{N}\mathbb{E}_{seeds}[\mathbb{E}[\mathbb{I}_{S_m=s}\mathbb{I}_{S_m=x}R_mV_t^{(TD)}(X_m')|seed]],$$

$$= \gamma\mathbb{E}_{seeds}\left[\sum_{k=1}^{N}\sum_{\substack{l=1,\\l\neq k}}^{N}\widehat{d_\pi}(s)\widehat{d_\pi}(x)\widehat{r_\pi}(s)(\widehat{\mathcal{P}_\pi}V_t^{(TD)})(x)\right] + \gamma\mathbb{I}_{s=x}N\mathbb{E}_{seeds}[\widehat{d_\pi}(s)r_\pi(s)(\widehat{\mathcal{P}_\pi}V_t^{(TD)})(s)],$$

$$\boxed{= \gamma N(N-1)d_\pi(s)d_\pi(x)r_\pi(s)(\mathcal{P}_\pi m_t^{(TD)})(x) + \gamma\mathbb{I}_{s=x}Nd_\pi(s)r_\pi(s)(\mathcal{P}_\pi m_t^{(TD)})(s)}$$

We will simplify Eq. (14) now.

$$\mathbb{E}_{seeds}\left[\left(\sum_{k=1}^{N}\mathbb{I}_{S_k=s}R_k\right)\left(\sum_{l=1}^{N}\mathbb{I}_{S_l=x}V_t^{(TD)}(x)\right)\right]$$

$$= \mathbb{E}_{seeds}\left[\sum_{k=1}^{N}\sum_{\substack{l=1,\\l\neq k}}^{N}\mathbb{I}_{S_k=s}\mathbb{I}_{S_l=x}R_kV_t^{(TD)}(x)\right] + \mathbb{E}_{seeds}\left[\sum_{m=1}^{N}\mathbb{I}_{S_m=s}\mathbb{I}_{S_m=x}R_mV_t^{(TD)}(x)\right]$$

$$= \mathbb{E}_{seeds}\left[\sum_{k=1}^{N}\sum_{\substack{l=1,\\l\neq k}}^{N}\mathbb{E}[\mathbb{I}_{S_k=s}\mathbb{I}_{S_l=x}R_kV_t^{(TD)}(x)|seed]\right] + \mathbb{I}_{s=x}\sum_{m=1}^{N}\mathbb{E}_{seeds}[\mathbb{E}[\mathbb{I}_{S_m=s}\mathbb{I}_{S_m=x}R_mV_t^{(TD)}(x)|seed]],$$

$$= \mathbb{E}_{seeds}\left[\sum_{k=1}^{N}\sum_{\substack{l=1,\\l\neq k}}^{N}\widehat{d_\pi}(s)\widehat{d_\pi}(x)\widehat{r_\pi}(s)V_t^{(TD)}(x)\right] + \mathbb{I}_{s=x}N\mathbb{E}_{seeds}[\widehat{d_\pi}(s)r_\pi(s)V_t^{(TD)}(s)],$$

$$\boxed{= N(N-1)d_\pi(s)d_\pi(x)r_\pi(s)m_t^{(TD)}(x) + \mathbb{I}_{s=x}Nd_\pi(s)r_\pi(s)m_t^{(TD)}(s)}$$

The next expression Eq. (15) is similar to Eq. (13) and therefore, we skip the simplification details.

$$\gamma\mathbb{E}_{seeds}\left[\left(\sum_{k=1}^{N}\mathbb{I}_{S_k=s}V_t^{(TD)}(S_k')\right)\left(\sum_{l=1}^{N}\mathbb{I}_{S_l=x}R_l\right)\right]$$

$$\boxed{= \gamma N(N-1)d_\pi(s)d_\pi(x)r_\pi(x)(\mathcal{P}_\pi m_t^{(TD)})(s) + \gamma\mathbb{I}_{s=x}Nd_\pi(s)r_\pi(s)(\mathcal{P}_\pi m_t^{(TD)})(s)}$$

Now, we simplify (16),

$$\gamma^2\mathbb{E}_{seeds}\left[\left(\sum_{k=1}^{N}\mathbb{I}_{S_k=s}V_t^{(TD)}(S_k')\right)\left(\sum_{l=1}^{N}\mathbb{I}_{S_l=x}V_t^{(TD)}(X_l')\right)\right]$$

$$= \gamma^2\mathbb{E}_{seeds}\left[\sum_{k=1}^{N}\sum_{\substack{l=1,\\l\neq k}}^{N}\mathbb{I}_{S_k=s}\mathbb{I}_{S_l=x}V_t^{(TD)}(S_k')V_t^{(TD)}(X_l')\right] + \gamma^2\mathbb{E}_{seeds}\left[\sum_{m=1}^{N}\mathbb{I}_{S_m=s}\mathbb{I}_{S_m=x}V_t^{(TD)}(S_m')V_t^{(TD)}(X_m')\right]$$

$$= \gamma^2\mathbb{E}_{seeds}\left[\sum_{k=1}^{N}\sum_{\substack{l=1,\\l\neq k}}^{N}\mathbb{E}[\mathbb{I}_{S_k=s}\mathbb{I}_{S_l=x}V_t^{(TD)}(S_k')V_t^{(TD)}(X_l')|seed]\right]$$
$$+ \gamma^2\mathbb{I}_{s=x}\sum_{m=1}^{N}\mathbb{E}_{seeds}[\mathbb{E}[\mathbb{I}_{S_m=s}\mathbb{I}_{S_m=x}V_t^{(TD)}(S_m')V_t^{(TD)}(X_m')|seed]],$$

$$= \gamma^2\mathbb{E}_{seeds}\left[\sum_{k=1}^{N}\sum_{\substack{l=1,\\l\neq k}}^{N}\widehat{d_\pi}(s)\widehat{d_\pi}(x)(\widehat{\mathcal{P}_\pi}V_t^{(TD)})(s)(\widehat{\mathcal{P}_\pi}V_t^{(TD)})(x)\right]$$
$$+ \gamma^2\mathbb{I}_{s=x}\sum_{m=1}^{N}\mathbb{E}_{seeds}[\widehat{d_\pi}(s)(\widehat{\mathcal{P}_\pi}V_t^{(TD)})(s)(\widehat{\mathcal{P}_\pi}V_t^{(TD)})(x)],$$

$$\boxed{= \gamma^2 N(N-1)d_\pi(s)d_\pi(x)(\mathcal{P}_\pi\Sigma_t^{(TD)}\mathcal{P}_\pi^T)(s,x) + \gamma^2\mathbb{I}_{s=x}Nd_\pi(s)(\mathcal{P}_\pi diag(\Sigma_t^{(TD)}))(s)}$$

We now consider the next piece Eq. (17),

$$\gamma\mathbb{E}_{seeds}\left[\left(\sum_{k=1}^{N}\mathbb{I}_{S_k=s}V_t^{(TD)}(S_k')\right)\left(\sum_{l=1}^{N}\mathbb{I}_{S_l=x}V_t^{(TD)}(x)\right)\right]$$

$$= \gamma \mathbb{E}_{seeds} \left[ \sum_{k=1}^{N} \sum_{\substack{l=1, \\ l \neq k}}^{N} \mathbb{I}_{S_k=s} \mathbb{I}_{S_l=x} V_t^{(TD)}(S_k') V_t^{(TD)}(x) \right] + \gamma \mathbb{E}_{seeds} \left[ \sum_{m=1}^{N} \mathbb{I}_{S_m=s} \mathbb{I}_{S_m=x} V_t^{(TD)}(S_m') V_t^{(TD)}(x) \right]$$

$$= \gamma \mathbb{E}_{seeds} \left[ \sum_{k=1}^{N} \sum_{\substack{l=1, \\ l \neq k}}^{N} \mathbb{E}[\mathbb{I}_{S_k=s} \mathbb{I}_{S_l=x} V_t^{(TD)}(S_k') V_t^{(TD)}(x) | seed] \right]$$

$$+ \gamma \mathbb{I}_{s=x} \sum_{m=1}^{N} \mathbb{E}_{seeds}[\mathbb{E}[\mathbb{I}_{S_m=s} \mathbb{I}_{S_m=x} V_t^{(TD)}(S_m') V_t^{(TD)}(x) | seed]],$$

$$= \gamma \mathbb{E}_{seeds} \left[ \sum_{k=1}^{N} \sum_{\substack{l=1, \\ l \neq k}}^{N} \widehat{d_\pi}(s) \widehat{d_\pi}(x) (\widehat{\mathcal{P}_\pi} V_t^{(TD)})(s) V_t^{(TD)}(x) \right] + \gamma \mathbb{I}_{s=x} \sum_{m=1}^{N} \mathbb{E}_{seeds}[\widehat{d_\pi}(s) \ (\widehat{\mathcal{P}_\pi} V_t^{(TD)})(s) V_t^{(TD)}(s)],$$

$$\boxed{= \gamma N(N-1) d_\pi(s) d_\pi(x) (\mathcal{P}_\pi \Sigma_t^{(TD)})(s,x) + \gamma \mathbb{I}_{s=x} N d_\pi(s) (\mathcal{P}_\pi \Sigma_t^{(TD)})(s,s).}$$

Simplification of Eq. (18) is similar to Eq. (14).

$$\mathbb{E}_{seeds} \left[ \left( \sum_{k=1}^{N} \mathbb{I}_{S_k=s} V_t^{(TD)}(s) \right) \left( \sum_{l=1}^{N} \mathbb{I}_{S_l=x} R_l \right) \right]$$

$$\boxed{= N(N-1) d_\pi(s) d_\pi(x) r_\pi(x) m^{(TD)}(s) + \mathbb{I}_{s=x} N d_\pi(s) r_\pi(s) m^{(TD)}(s).}$$

Also, the simplification of Eq. (19) is similar to that of Eq. (17).

$$\gamma \mathbb{E}_{seeds} \left[ \left( \sum_{k=1}^{N} \mathbb{I}_{S_k=s} V_t^{(TD)}(s) \right) \left( \sum_{l=1}^{N} \mathbb{I}_{S_l=x} V_t^{(TD)}(X_l') \right) \right]$$

$$\boxed{= \gamma N(N-1) d_\pi(s) d_\pi(x) (\mathcal{P}_\pi \Sigma_t^{(TD)})^T(s,x) + \gamma \mathbb{I}_{s=x} N d_\pi(s) (\mathcal{P}_\pi \Sigma_t^{(TD)})^T(s,s).}$$

We will simplify the final part i.e., Eq. (20).

$$\mathbb{E}_{seeds} \left[ \left( \sum_{k=1}^{N} \mathbb{I}_{S_k=s} V_t^{(TD)}(s) \right) \left( \sum_{l=1}^{N} \mathbb{I}_{S_l=x} V_t^{(TD)}(x) \right) \right]$$

$$= \mathbb{E}_{seeds} \left[ \sum_{k=1}^{N} \sum_{\substack{l=1, \\ l \neq k}}^{N} \mathbb{I}_{S_k=s} \mathbb{I}_{S_l=x} V_t^{(TD)}(s) V_t^{(TD)}(x) \right] + \mathbb{E}_{seeds} \left[ \sum_{m=1}^{N} \mathbb{I}_{S_m=s} \mathbb{I}_{S_m=x} V_t^{(TD)}(s) V_t^{(TD)}(x) \right]$$

$$= \mathbb{E}_{seeds} \left[ \sum_{k=1}^{N} \sum_{\substack{l=1, \\ l \neq k}}^{N} \mathbb{E}[\mathbb{I}_{S_k=s} \mathbb{I}_{S_l=x} V_t^{(TD)}(s) V_t^{(TD)}(x) | seed] \right]$$

$$+ \gamma \mathbb{I}_{s=x} \sum_{m=1}^{N} \mathbb{E}_{seeds}[\mathbb{E}[\mathbb{I}_{S_m=s} \mathbb{I}_{S_m=x} V_t^{(TD)}(s) V_t^{(TD)}(x) | seed]],$$

$$= \mathbb{E}_{seeds} \left[ \sum_{k=1}^{N} \sum_{\substack{l=1, \\ l \neq k}}^{N} \widehat{d_\pi}(s) \widehat{d_\pi}(x) V_t^{(TD)}(s) V_t^{(TD)}(x) \right] + \gamma \mathbb{I}_{s=x} \sum_{m=1}^{N} \mathbb{E}_{seeds}[\widehat{d_\pi}(s) V_t^{(TD)}(s) V_t^{(TD)}(s)],$$

$$\boxed{= N(N-1) d_\pi(s) d_\pi(x) \Sigma_t^{(TD)}(s,x) + \gamma \mathbb{I}_{s=x} N d_\pi(s) \Sigma_t^{(TD)}(s,s).}$$

We can put all these pieces together to get the desired result.

$$
\begin{aligned}
\Sigma_t^{(TD)}(s,x) &= \Sigma_t^{(TD)}(s,x) + \alpha\Omega_t^{(TD)}(s,x) + \alpha^2\Psi_t^{(TD)}(s,x), \\
\Omega_t^{(TD)}(s,x) &= Nd_\pi(x)\left(r_\pi(x)m_t^{(TD)}(s) + \gamma(\mathcal{P}_\pi\Sigma_t^{(TD)})(x,s) - \Sigma_t^{(TD)}(s,x)\right) \\
&\quad + Nd_\pi(s)\left(r_\pi(s)m_t^{(TD)}(x) + \gamma(\mathcal{P}_\pi\Sigma_t^{(TD)})(s,x) - \Sigma_t^{(TD)}(x,s)\right), \\
\Psi_t^{(TD)}(s,x) &= N(N-1)d_\pi(s)d_\pi(x)r_\pi(s)r_\pi(x) + \mathbb{I}_{s=x}Nd_\pi(s)\left(c(s) + r_\pi^2(s)\right) \\
&\quad + \gamma N(N-1)d_\pi(s)d_\pi(x)r_\pi(s)(\mathcal{P}_\pi m_t^{(TD)})(x) + \gamma\mathbb{I}_{s=x}Nd_\pi(s)r_\pi(s)(\mathcal{P}_\pi m_t^{(TD)})(s) \\
&\quad - N(N-1)d_\pi(s)d_\pi(x)r_\pi(s)m_t^{(TD)}(x) - \mathbb{I}_{s=x}Nd_\pi(s)r_\pi(s)m_t^{(TD)}(s) \\
&\quad + \gamma N(N-1)d_\pi(s)d_\pi(x)r_\pi(x)(\mathcal{P}_\pi m_t^{(TD)})(s) + \gamma\mathbb{I}_{s=x}Nd_\pi(s)r_\pi(s)(\mathcal{P}_\pi m_t^{(TD)})(s) \\
&\quad + \gamma^2 N(N-1)d_\pi(s)d_\pi(x)(\mathcal{P}_\pi\Sigma_t^{(TD)}\mathcal{P}_\pi^T)(s,x) + \gamma^2\mathbb{I}_{s=x}Nd_\pi(s)(\mathcal{P}_\pi diag(\Sigma_t^{(TD)}))(s) \\
&\quad - \gamma N(N-1)d_\pi(s)d_\pi(x)(\mathcal{P}_\pi\Sigma_t^{(TD)})(s,x) - \gamma\mathbb{I}_{s=x}Nd_\pi(s)(\mathcal{P}_\pi\Sigma_t^{(TD)})(s,s) \\
&\quad - N(N-1)d_\pi(s)d_\pi(x)r_\pi(x)m_t^{(TD)}(s) - \mathbb{I}_{s=x}Nd_\pi(s)r_\pi(s)m_t^{(TD)}(s) \\
&\quad - \gamma N(N-1)d_\pi(s)d_\pi(x)(\mathcal{P}_\pi\Sigma_t^{(TD)})^T(s,x) - \gamma\mathbb{I}_{s=x}Nd_\pi(s)(\mathcal{P}_\pi\Sigma_t^{(TD)})^T(s,s) \\
&\quad + N(N-1)d_\pi(s)d_\pi(x)\Sigma_t^{(TD)}(s,x) + \mathbb{I}_{s=x}Nd_\pi(s)\Sigma_t^{(TD)}(s,s)
\end{aligned}
$$

We will now prove the final part of the theorem. The mean squared error $\xi_{t+1}^{(TD)}$ can be decomposed into bias and covariance terms:

$$
\begin{aligned}
\xi_{t+1}^{(TD)} &= \sum_s (bias_{t+1}^2(s) + Cov_{t+1}(s,s)), \\
&= \sum_s \left((v_\pi(s) - m_{t+1}^{(TD)}(s)) + (\Sigma_{t+1}^{(TD)}(s,s) - m_{t+1}^{(TD)}(s)m_{t+1}^{(TD)}(s))\right), \\
&= \sum_s (v_\pi^2(s) - 2v_\pi(s)m_{t+1}^{(TD)}(s) + \Sigma_{t+1}^{(TD)}(s,s)), \\
&= \sum_s \left(v_\pi^2(s) - 2v_\pi(s)(m_t^{(TD)}(s) + \alpha\Delta_t^{(TD)}(s)) + \Sigma_t^{(TD)}(s,s) + \alpha\Omega_t^{(TD)}(s,s) + \alpha^2\Psi_t^{(TD)}(s,s)\right), \\
&= \sum_s \left(v_\pi^2(s) - 2v_\pi(s)m_t^{(TD)}(s) + \Sigma_t^{(TD)}(s,s)\right) \\
&\qquad + \sum_s \left(-2\alpha v_\pi(s)\Delta_t^{(TD)}(s)) + \alpha\Omega_t^{(TD)}(s,s) + \alpha^2\Psi_t^{(TD)}(s,s)\right), \\
\boxed{&= \xi_t^{(TD)} + \alpha\sum_{s\in\mathcal{S}}(-2v_\pi(s)\Delta_t^{(TD)}(s) + \Omega_t^{(TD)}(s,s)) + \alpha^2\sum_{s\in\mathcal{S}}\Psi_t^{(TD)}(s,s).}
\end{aligned}
$$

$\square$

**Theorem B.3.4.** *Let* $\mathbb{E}_{seeds}$ *denote the expectation with respect to seeds. Let*

$$
m_t^{(PT)}(s) = \mathbb{E}_{seeds}[V_t^{(PT)}(s)],
$$
$$
Cov(V_t^{(PT)}(s), V_t^{(PT)}(x)) = \Sigma_t^{(PT)}(s,x) - m_t^{(PT)}(s)m_t^{(PT)}(x),
$$

*where* $\Sigma_t^{(PT)}(s,x) = \mathbb{E}_{seeds}[V_t^{(PT)}(s)V_t^{(PT)}(s)]$. *Then:*

$$
m_{t+1}^{(PT)}(s) = m_t^{(PT)}(s) + \alpha Nd_\pi(s)\Delta_t^{(PT)}(s),
$$
$$
\Sigma_{t+1}^{(PT)}(s,x) = \Sigma_t^{(PT)}(s,x) + \alpha\Omega_t^{(PT)}(s,x) + \alpha^2\Psi_t^{(PT)}(s,x),
$$

*and, the mean squared error $\xi_t^{(PT)}$ at timestep $t$ is*

$$\xi_{t+1}^{(PT)} = \xi_t^{(PT)} + \alpha \sum_{s \in \mathcal{S}} d_\pi(s)(-2v_\pi(s)\Delta_t^{(PT)}(s) + \Omega_t^{(PT)}(s,s))$$

$$+ \alpha^2 \sum_{s \in \mathcal{S}} d_\pi(s)\Psi_t^{(PT)}(s,s).$$

*Proof.* The proof is similar to that of TD learning algorithm. We have:

$$m_{t+1}^{(PT)}(s) = m_t^{(PT)}(s) + \alpha N d_\pi(s)(r_\pi(s) + \gamma(\mathcal{P}_\pi m_t^{(PT)})(s) - m_t^{(PT)}(s)),$$

$$\Sigma_t^{(PT)}(s,x) = \Sigma_t^{(PT)}(s,x) + \alpha \Omega_t^{(PT)}(s,x) + \alpha^2 \Psi_t^{(PT)}(s,x),$$

$$\Omega_t^{(PT)}(s,x) = N d_\pi(x)\left(r_\pi(x)m_t^{(PT)}(s) + \gamma(\mathcal{P}_\pi\Sigma_t^{(PT)})(x,s) - \Sigma_t^{(PT)}(s,x)\right)$$

$$+ N d_\pi(s)\left(r_\pi(s)m_t^{(PT)}(x) + \gamma(\mathcal{P}_\pi\Sigma_t^{(PT)})(s,x) - \Sigma_t^{(PT)}(x,s)\right),$$

$$\Psi_t^{(PT)}(s,x) = N(N-1)d_\pi(s)d_\pi(x)r_\pi(s)r_\pi(x) + \mathbb{I}_{s=x}N d_\pi(s)\left(c(s) + r_\pi^2(s)\right)$$

$$+ \gamma N(N-1)d_\pi(s)d_\pi(x)r_\pi(s)(\mathcal{P}_\pi m_t^{(PT)})(x) + \gamma\mathbb{I}_{s=x}N d_\pi(s)r_\pi(s)(\mathcal{P}_\pi m_t^{(PT)})(s)$$

$$- N(N-1)d_\pi(s)d_\pi(x)r_\pi(s)m_t^{(PT)}(x) - \mathbb{I}_{s=x}N d_\pi(s)r_\pi(s)m_t^{(PT)}(s)$$

$$+ \gamma N(N-1)d_\pi(s)d_\pi(x)r_\pi(x)(\mathcal{P}_\pi m_t^{(PT)})(s) + \gamma\mathbb{I}_{s=x}N d_\pi(s)r_\pi(s)(\mathcal{P}_\pi m_t^{(PT)})(s)$$

$$+ \gamma^2 N(N-1)d_\pi(s)d_\pi(x)(\mathcal{P}_\pi\Sigma_t^{(PT)}\mathcal{P}_\pi^T)(s,x) + \gamma^2\mathbb{I}_{s=x}N d_\pi(s)(\mathcal{P}_\pi diag(\Sigma_t^{(PT)}))(s)$$

$$- \gamma N(N-1)d_\pi(s)d_\pi(x)(\mathcal{P}_\pi\Sigma_t^{(PT)})(s,x) - \gamma\mathbb{I}_{s=x}N d_\pi(s)(\mathcal{P}_\pi\Sigma_t^{(PT)})(s,s)$$

$$- N(N-1)d_\pi(s)d_\pi(x)r_\pi(x)m_t^{(PT)}(s) - \mathbb{I}_{s=x}N d_\pi(s)r_\pi(s)m_t^{(PT)}(s)$$

$$- \gamma N(N-1)d_\pi(s)d_\pi(x)(\mathcal{P}_\pi\Sigma_t^{(PT)})^T(s,x) - \gamma\mathbb{I}_{s=x}N d_\pi(s)(\mathcal{P}_\pi\Sigma_t^{(PT)})^T(s,s)$$

$$+ N(N-1)d_\pi(s)d_\pi(x)\Sigma_t^{(PT)}(s,x) + \mathbb{I}_{s=x}N d_\pi(s)\Sigma_t^{(PT)}(s,s)$$

The MSE recursive equation can be obtained similar to that of TD expression. Therefore, we skip the details and give the final expression.

$$\xi_{t+1}^{(PT)} = \xi_t^{(PT)} + \alpha \sum_{s \in \mathcal{S}}(-2v_\pi(s)\Delta_t^{(PT)}(s) + \Omega_t^{(PT)}(s,s)) + \alpha^2 \sum_{s \in \mathcal{S}}\Psi_t^{(PT)}(s,s).$$

$\square$

# C   Experiments

Besides providing details for the experiments described in the main paper, we answer additional interesting questions, such as:

- What is the effect of task distribution on the performance for various methods (Sec. C.2)?
- What is the effect of network capacity on the performance (Sec. C.3)?
- What is the effect of hyperparameters on the performance (Sec. C.4)?

## C.1   Additional Details

We begin by providing details of the experiments.

### C.1.1   Policy evaluation grids

We modify the rewards of the goal states as described in Table C.1. The true value function for each task is shown in Fig. C.1. We convert the 5x5 discrete grid into a 48x48 RGB image for the deep RL

| Task ID | Reward | | | |
|---|---|---|---|---|
| | Top-Left | Top-Right | Bottom-Left | Bottom-Right |
| 1 | 0 | 1 | 0 | 1 |
| 2 | 1 | 0 | 1 | 0 |
| 3 | 0 | 0 | 1 | 1 |
| 4 | 1 | 1 | 0 | 0 |

Table C.1: Rewards at goal states for various tasks.

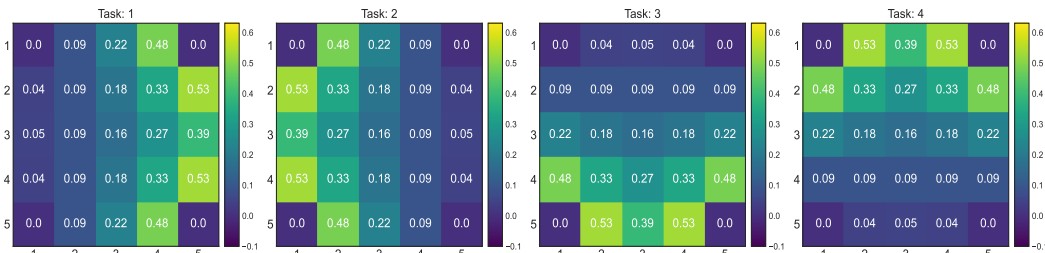

Figure C.1: Heatmap of the true value function for different tasks.

task. The agent's location is indicated by colouring the its location in red and we use green squares to indicate goal states. At every timestep, the agent receives a RGB image as an input. The action space and the action consequences are the same as discrete grid. We select best hyperparameters (learning rates) for each algorithm by choosing those values which results in the lowest area under the error curves as shown in Table C.6 and Fig. C.9. The hyperparameter search and the final reporting is performed by averaging the results on 30 different seeds. For our method, the buffer stores the samples from the latest task only to update the permanent value function.

For the first linear FA experiment, we use the same setup as the tabular setting except we encode each state in the grid as a vector of 10 bits. The first 5 bits encodes the row of the state and the last 5 bits encodes the column of the state. For example, the starting state (2,2) is represented as [0,0,1,0,0,0,0,1,0,0]. Both permanent value function and transient value function use same features to learn respective estimates.

For the second linear FA experiment, we use a continuous version of the grid. The goal rewards for each task is same as the discrete counterpart as described in Table C.1. The true value function is also different for each task and it is shown as a heatmap in Fig. C.2. We use RBFs to generate features as described in Sutton and Barto [45]. We use 676 centers (order$^2$) spaced equally in both x and y directions across the grid with a variance $(0.75/(order - 1))^2$ and use a threshold to convert the continuous vector into a binary feature vector. Any feature value greater than 0.5 is set to 1, while those features with values less than 0.5 is set to 0. We search for the best learning rate from the same set of values as the discrete grid task. The hyperparameter curves are shown in Fig. C.11.

For the deep RL experiment, the rewards are multiplied by a factor of 10 to minimize the initialization effect of deep neural network on prediction. We used a convolution neural network (CNN) to approximate the value function for all algorithms tested. The CNN consists of two conv layers with

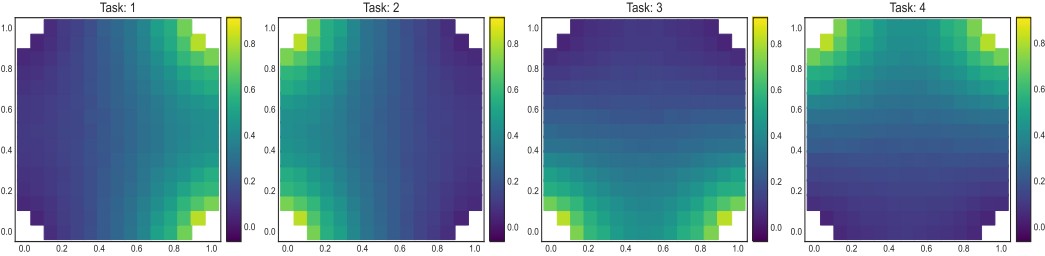

Figure C.2: Heatmap of the true value function for different tasks.

8 filters, (2,2) stride, and relu activation in each layer. We use a max pooling layer in between the two conv layers with a stride 2. The output of the second conv layer is flatted, and it is followed by a couple of linear layers. The fully connected network has 64 and 1 hidden units in the respective layers. The first fully connected layer uses a tanh activation and no activation is used in the final layer. For our method, we attach a new head after the first two conv layers for permanent value function. This head also has two linear layers where the first layer has tanh activation. The entire network is trained by backpropagating the errors from the transient value function. But, the conv layers are not trained when updating the permanent value function. The transient value function and the TD learning algorithm are trained in an online fashion using an experience replay buffer of size 100k. The buffer is reset at the end of the task for all methods to reduce the bias from old samples. permanent value function is trained at the end of the task using the samples from the latest task. 100 gradient steps are taken during each update cycle. We use SGD optimizer to update its weights. The best learning rate is picked by computing the area under the error curve on a set of values shown in Table C.7. The mean AUC curves of various choices of learning rates are shown in Fig. C.12.

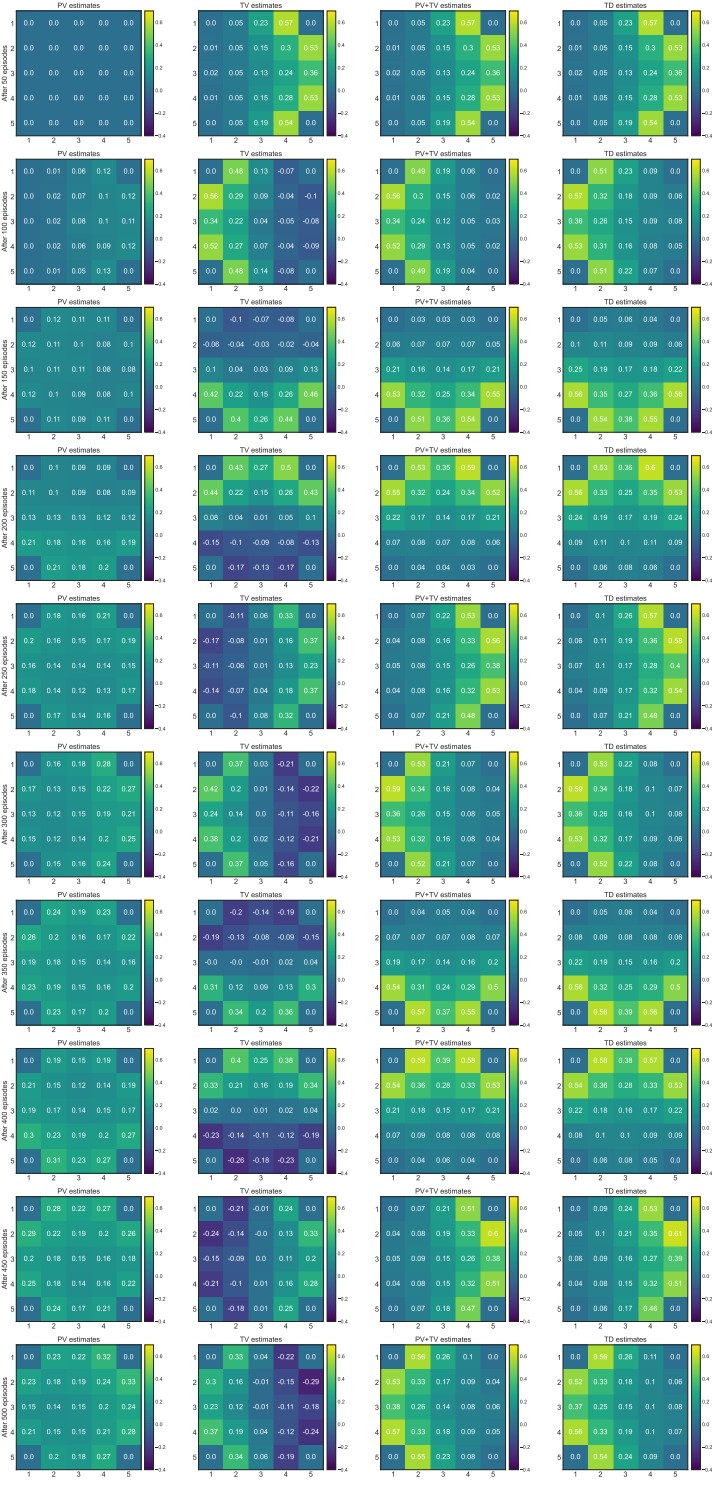

Figure C.3: Predictions evolution across episodes.

## C.1.2 Policy evaluation minigrid

We use 4 rooms task as shown in Fig. 1b. The environment is a grid task with each room containing one type of item. Each item type is different in terms of how it is perceived and in terms of reward the agent receives upon collecting it. There are 4 goal cells, one in each room located at the corner. At

the beginning of the episode, the agent starts in one of the cells within the region spanning between cells (3,3) and (6,6). It has 3 actions to chose from turn left, turn right, and move forward. Turn-based actions changes the orientation of the agent along the corresponding direction, whereas the move action changes the agent's location to the state in front of it. The agent can only observe a 5x5 view in front of it (partial view). Items are one-hot represented along the 3rd dimension to make the overall input space 5x5x5. To set up the continual learning problem, we change the rewards of the items from task-to-task as described in Table C.2. The evaluation policy is kept uniformly random for all states and for all the tasks. We use a discount factor of 0.9. We evaluate the performance for a total of 750 episodes and the tasks are changes after every 75 episodes.

| Task ID | Reward | | | |
| | Blue (●) | Red (●) | Yellow(●) | Purple (●) |
| --- | --- | --- | --- | --- |
| 1 | -1 | 1 | 1 | 1 |
| 2 | 1 | 1 | 0.5 | 1 |
| 3 | 1 | 1 | 1 | -1 |
| 4 | -1 | 1 | 0.5 | -1 |

Table C.2: Rewards for items for various tasks.

We used a convolution neural network (CNN) to approximate the value function for all algorithm. The CNN consists of three conv layers with 16, 32, and 64 filters, (2,2) stride and relu activation in each layer. The output of the third conv layer is flatted, which is then followed by a couple of linear layers with 64 and 1 hidden unit. Tanh activation in the first linear layer and no activation is used in the last layer. We attach a head after conv layers for permanent value function. The architecture of this head is same as the TD learning head. The training method and buffer capacities are same as the discrete image grid task. We use SGD optimizer to update weights in all networks. The best learning rate is picked by computing the area under the error curve on a set of values shown in Table C.8 and Fig. C.13. All the results reported are averaged across 30 seeds.

### C.1.3 Control minigrid

We use two room environment as shown in Fig. 3c. The agent starts in one of the bottom row states in the bottom room and there are two goal states located in the top room. In the first task, the goal rewards are +5 and 0 for the first and second goals respectively, and it is flipped for the second task. We alternate between two tasks every 500 episodes for a total of 2500 episodes. The agent receives a one hot, 5x5 view (partial observability) in front of it at each timestep. It can take one of the three actions (move forward, turn left, turn right) at every step. The *move* action transitions the agent to the state in front of it if there is no obstacle. The environment is deterministic and episodic, and we use a discount factor of 0.99.

We use a CNN as a function approximator. The CNN has 3 conv layers with 16, 32, and 64 filters, (2,2) stride, and relu activations. It is followed by a fully connected neural network with 128, 64, and 4 hidden units. The first layer is relu activated, the second layer is tanh activated, and the final layer is linear. We use target network and experience replay to train the network. Target network's weights are updated every 200 timesteps. Experience replay buffer's capacity is capped to 100k and the stored samples are deleted when the task changes. We use batch size of 64 and update the network every timestep. We search for the best learning rate from a set of value shown in Table C.10 on 3 seeds and we pick the one that gave the highest area under the rewards curve. We then run on 30 seeds using that learning rate to report the final performance. The transient value function training procedure and details are identical to DQN. permanent value function uses a separate head following conv layers and is updated using Adam optimizer whenever the task is changed.

### C.1.4 PT-DQN Pseudocode

---
**Algorithm 4** PT-DQN Pseudocode (Continual Reinforcement Learning)

---
1: Initialize transient network buffer $\mathcal{B}$, permanent network buffer $\mathcal{D}$
2: Initialize permanent network $\theta$, transient network $\mathbf{w}$
3: Initialize target network $\mathbf{w}_{target}$
4: **for** $t : 0 \to \infty$ **do**
5:     Take action $A_t$ according to $\epsilon$-greedy policy
6:     Observe $R_{t+1}, S_{t+1}$
7:     Store $(S_t, A_t, R_{t+1}, S_{t+1})$ in $\mathcal{B}$
8:     Store $(S_t, A_t, Q^{(P)}(S_t, A_t))$ in $\mathcal{D}$
9:     Sample a batch of transitions $(S_j, A_j, R_{j+1}, S_{j+1})$ from $B$
10:    Compute target
$$y = R_{j+1} + \gamma \max_{a' \in \mathcal{A}} \left( Q^{(P)}(S_{j+1}, a'; \theta) + Q^{(T)}(S_{j+1}, a'; \mathbf{w}_{target}) \right) - Q^{(P)}(S_j, A_j; \theta)$$
11:    Perform a gradient step on $\left( y - Q^{(T)}(S_j, A_j; \mathbf{w}) \right)^2$ with respect to $\mathbf{w}$
12:    Update target network every $N$ steps, $\mathbf{w}_{target} = \mathbf{w}$
13:    **if** $mod(t, k) == 0$ **then**
14:       Sample a batch of transitions $(S_j, A_j, Q^{(P)}(S_j, A_j))$ from $D$
15:       Compute target $\hat{y} = Q^{(P)}(S_j, A_j) + Q^{(T)}(S_j, A_j; \mathbf{w})$
16:       Perform a gradient step on $\left( \hat{y} - Q^{(P)}(S_j, A_j; \theta) \right)^2$ with respect to $\theta$
17:       Decay transient network weights, $\mathbf{w} = \lambda \mathbf{w}$
18:       Clear permanent network buffer, $D = \{\}$
19:    **end if**
20: **end for**

---

### C.1.5 JellyBeanWorld (JBW)

| Item | Blue (●) | Green (●) | Red (●) |
|---|---|---|---|
| Color | [0, 0, 1.0] | [0, 1.0, 0] | [1.0, 0, 0.0] |
| Intensity | Constant [-2.0] | Constant [-6.0] | Constant [-2.0] |
| ● | Piecewise box [2, 100, 10, -5] | Zero | Piecewise box [150, 0, -100, -100] |
| Interactions ● | Zero | Zero | Zero |
| ● | Piecewise box [150, 0, -100, -100] | Zero | Piecewise box [2, 100, 10, -5] |

Table C.3: JellyBeanWorld environment parameters.

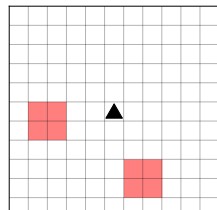

(a) Sample observation received by the agent.

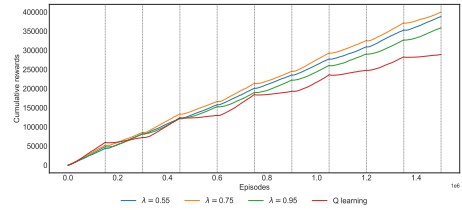

(b) Cummulative rewards against timesteps averaged across 3 seeds.

Figure C.4: JellyBeanWorld sample observation and performance curves for various decay values.

We test algorithms on a continual learning problem in the JBW testbed [31] and use neural networks as the function approximator. The environment is a two-dimensional infinite grid with three types of items: blue, red, and green. We induce spatial non-stationarity by setting the parameters of the environment as described in Table C.3 . In our configuration, three to four similar items (red or blue) form a tiny group and similar groups appear close to each other to form a sea of red (or blue) objects as shown in Fig. 5a. Therefore, the local observation distribution at various parts of the environment are different. At each timestep, the agent receives a egocentric 11x11 RGB, $360^o$ view

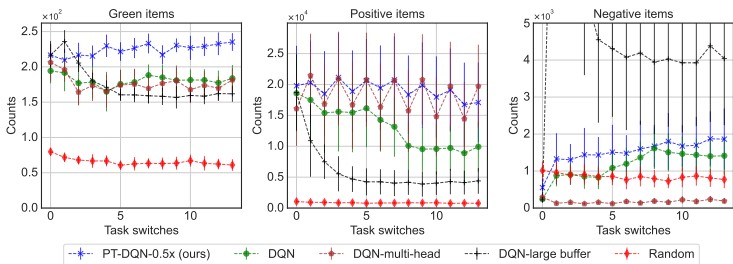

Figure C.5: Items collected by various algorithms during each task switches.

as an observation as shown in Fig. C.4a. It has four actions to choose from: up, down, left, and right. Each action transitions it by one square along the corresponding direction. The rewards for green items, which are uniformly distributed, are set to 0.1, whereas the rewards for picking red and blue items alternate between -1 and +2 every 150k timesteps, therefore, inducing reward non-stationarity. We run the experiment for 2.1 million timesteps and train algorithms using a discount factor 0.9.

The DQN agent uses four layered neural network with relu activations in all layers except the last to estimate q-values. The number of hidden units are 512, 256, 128, and 4 respectively. The network is trained using Adam optimizer with experience replay and a target network. We use epsilon-greedy policy with $\epsilon = 0.1$ for exploration. The permanent and transient network architectures are identical to that of DQN but with half the number of parameters in each layer. This ensures that the total number of parameters for the baselines and our method is same. Transient network is also trained using Adam optimizer, whereas, permanent network is updated every 10k timesteps ($k$=10,000) using SGD optimizer. Target network's weights are updated every 200 timesteps. Experience replay buffer capacity is capped to 8000 as higher capacity buffer stores old training samples which affects the training when the environment is changing continuously. We use batch size of 64 and update the network every timestep. For our approach, we experimented with three choices of $\lambda$ values (0.55, 0.75, and 0.95) and all the choices resulted in a better performance than the DQN agent as seen in Fig. C.4b. We use the same exploration policy as the DQN agent. We flatten the observation into one long array, then stack the last four observation arrays to form an input to the neural network. The best learning rate for all algorithms are chosen by comparing the total reward obtained in 1.5 million timesteps for a set of fixed learning rate values on 3 different seeds. We search for the best learning rate from a set of value shown in Table C.12 on 3 seeds and we pick the one that gave the highest rewards. We then run on 30 seeds using that learning rate to report the final performance. The random agent picks one of the four actions according to uniform probability at each timestep. For the multi-head DQN baseline, we attach two output heads - one for each task. The baseline knows the task IDs, so it chooses the appropriate head to select actions and to train the network. The replay buffer is reset for this baseline whenever the task is changed. We use a buffer capacity of 250k for the DQN baseline that is augmented with a large buffer. The rest of the training and architecture details are same as the DQN agent.

### C.1.6 MinAtar

This domain is a standard benchmark for testing single task RL algorithms [51]. We use breakout, freeway, and space invaders environments to setup a continual RL problem by randomly picking one of these tasks every 500k steps for a total of 3.5M steps. We standardize the observation space across tasks by padding 0s when needed and make all the actions available for the agent. This problem has non-stationarity in transitions besides the reward and observation signals making it challenging. We use a discount factor of 0.99 for this experiment.

We used a CNN which has 1 relu activated conv layer with 16 filters followed by a relu activated linear layer with 256 units, and a linear output layer for DQN. The architecture is same as the one used in the original paper [51]. The architecture is same for other DQN variants but we add 3 output layers for the multi-headed DQN. For our method, we use the same architecture but 8 filter in the conv layer and 128 units in the linear layer. Target network is used for all the methods which is updated every 1000 steps and we use an experience replay of 100k for all methods but we use 500k sized experience replay for large replay based DQN baseline. We use Adam optimizer to train the networks in online fashion. For our method, permanent value function network is updated using SGD

optimizer every 50k steps. The best hyperparameters are chosen by running the experiment for 1.5M steps by switching the tasks every 500k steps on 3 seeds. The results of hyperparameter tuning are detailed in Tables C.17, C.18, C.19, and C.20. We make all 7 actions available to the agent at all times and the input to the agent is 10x10x7 images.

## C.2 What is the effect of task distribution on the performance for various methods?

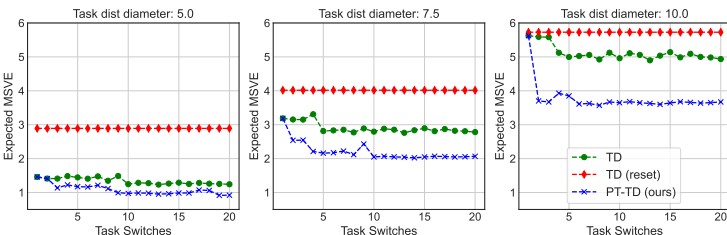

Figure C.6: Expected analytic MSVE on a task for various algorithms and $\epsilon$ values.

**Setup:** We set up a continual learning problem by creating 7 tasks whose value functions are $\epsilon$ (task diameter) distance from one another in norm-2. Each task has 7 states. For a particular $\epsilon$ value, for a particular number of task switches, we run each algorithm 100 times and save the final estimates which form the initial point for the new task. We then sample a task according to uniform distribution and use analytical MSE expression derived in the previous section to analyze the behaviour of PT-TD learning and the two versions of TD learning algorithm for 100 rounds. We then average the error across rounds and across different starting values to get a point in the plot shown in Fig. C.6.

**Task details:** We use the following $\mathcal{P}_\pi$ for all tasks and for all task diameters ($\epsilon$):

$$\mathcal{P}_\pi = \begin{bmatrix} 0.18 & 0.19 & 0.08 & 0.16 & 0.1 & 0.11 & 0.18 \\ 0.04 & 0.05 & 0.01 & 0.0 & 0.4 & 0.0 & 0.51 \\ 0.0 & 0.2 & 0.0 & 0.05 & 0.28 & 0.06 & 0.41 \\ 0.17 & 0.14 & 0.09 & 0.26 & 0.15 & 0.19 & 0.01 \\ 0.16 & 0.16 & 0.16 & 0.24 & 0.17 & 0.0 & 0.11 \\ 0.25 & 0.02 & 0.24 & 0.24 & 0.08 & 0.05 & 0.11 \\ 0.0 & 0.25 & 0.19 & 0.44 & 0.05 & 0.0 & 0.07 \end{bmatrix}$$

We then sample 7 points in $\mathbb{R}^n$ that are $\epsilon$ units from one another. These points form the true value function of the tasks. We then compute the reward vector for each task $\tau$ as $r_\pi(\tau) = (I - \gamma \mathcal{P}_\pi) v_\pi(\tau)$. At the time of testing the algorithms, we add a zero mean Gaussian noise with 0.1 standard deviation to these rewards to simulate stochasticity. We search from the learning rates shown in Table C.4 to pick the best one for each combination of $\epsilon$ and number of task switches. We use a learning rate of 0.01 while using analytical expressions.

**Observation:** We observe both TD learning algorithm and our approach has the same performance after seeing data from one task. But the error gap increases as the number of task switches increase. The low error in our method can be attributed to the presence of permanent value function which captures some portion of the value that is common for all tasks. transient value function then learns the remaining part with a small number of samples resulting in low error overall. We also observe that the error gap between TD learning and our method gets wider as we increase task diameter, which demonstrates that our method is broadly applicable. The reset variant of the TD learning algorithm whose predictions are reset whenever task changes has the largest error emphasizing learning from scratch could be costly.

| Algorithm | LRs |
|---|---|
| TD (without reset) | {8e-1 5e-1 3e-1 1e-1 5e-2 1e-2} |
| PT-TD | PV: {5e-1 3e-1 1e-1 5e-2 1e-2 5e-3} |
| | TV: {8e-1 5e-1 3e-1 1e-1 5e-2 1e-2} |

Table C.4: LRs considered to pick the best one.

## C.3 What is the effect of network capacity on the performance?

**Setup:** We use JBW task in the continual RL setting to perform the ablation study. We run two additional experiments by varying the number of parameters in DNN for the DQN and our method. In the first experiment, we use the same DNN architecture for all algorithms, therefore, our method has twice the number of parameters as that of DQN. In the second experiment, we double the network capacity for the DQN variant making the total number of parameters same as ours. We keep the rest of the experimental setup same as before and we pick the best learning rate based on the AUC by searching over the values detailed in Table C.5 over 3 seeds for 1.5M steps.

| Algorithm | LRs |
|-----------|-----|
| DQN (2x) | {1e-2, 1e-3, 1e-4, 1e-5, 1e-6, 1e-7} |
| PT-DQN | PV: {5e-1 3e-1 1e-1 5e-2 1e-2 5e-3} |
| | TV: {8e-1 5e-1 3e-1 1e-1 5e-2 1e-2} |

Table C.5: Search space of learning rates.

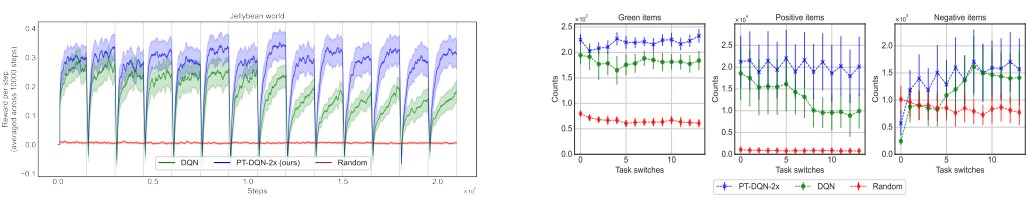

(a) Results on the JellyBeanWorld.  (b) Items collected in each task.

Figure C.7: (a) Rewards accumulated per timestep across 10k step window is plotted against timesteps. (b) Number of items collected within a task for each type (subplots use different scales along y-axis).

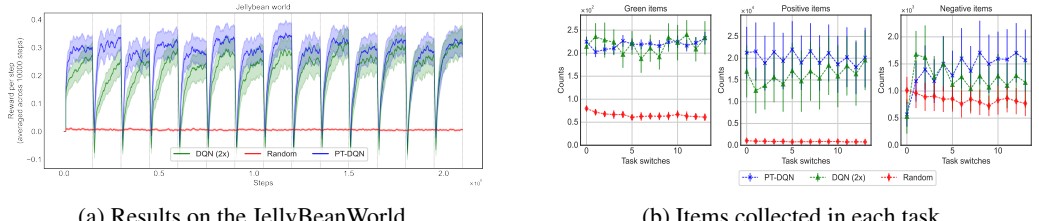

(a) Results on the JellyBeanWorld.  (b) Items collected in each task.

Figure C.8: (a) Rewards accumulated per timestep across 10k step window is plotted against timesteps. (b) Number of items collected within a task for each type (subplots use different scales along y-axis).

**Observations:** The results for ablation experiments are shown in Figures C.7 and C.8. We observe that our algorithm continues to perform better than the DQN agent in both the cases. The results also indicate that it is efficient to devote the available capacity in learning parts of the value function rather than learning the whole.

In the second ablation experiment, the DQN agent with twice the number of parameters starts off learning slowly but catches up with our method after seeing enough data. In fact, the DQN agent continues to perform similar to our method without any drop in performance as we continue to train further (we tested it by running the agent to 6M timesteps). This result highlights that our approach is beneficial in the *big world - small agent* setup where the computation budget of the agent is very small compared to the complexity of the environment. When the agent's capacity is large relative to the complexity of the environment, there's no additional benefit (neither there is any downside) to our method. Since the world is much more complicated in comparison to the agent's capacity, using our method is preferable.

These experiments also provide partial answer to the question - when does the plasticity problem arise in neural networks? We think that a combination of learning algorithm (like DQN), small agent

capacity, and a complex environment could give rise to the plasticity problem in neural networks. A thorough investigation is necessary to confirm this hypothesis.

## C.4 What is the effect of hyperparameters on the performance?

In this section we analyze the sensitivity of additional hyperparameter in our method (learning rate to update permanent value function) and compare it with the sensitivity of TD and Q-learning algorithms on various environments.

| Algorithm | LRs |
|---|---|
| TD (without reset) | {8e-1, 5e-1, 3e-1, 1e-1, 5e-2, 1e-2} |
| TD (with reset) | {8e-1, 5e-1, 3e-1, 1e-1, 5e-2, 1e-2} |
| PT-TD | PV: {1e-1, 5e-2, 1e-2, 5e-3, 1e-3} |
| | TV: {8e-1, 5e-1, 3e-1, 1e-1, 5e-2, 1e-2} |

Table C.6: LRs considered in the search space for the policy evaluation grid tasks.

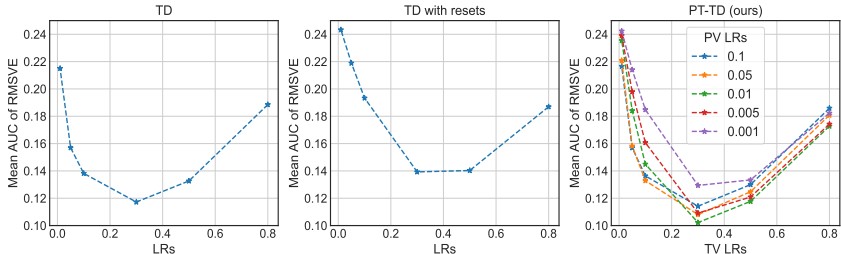

Figure C.9: Mean AUC of RMSVE against learning rate for various algorithms.

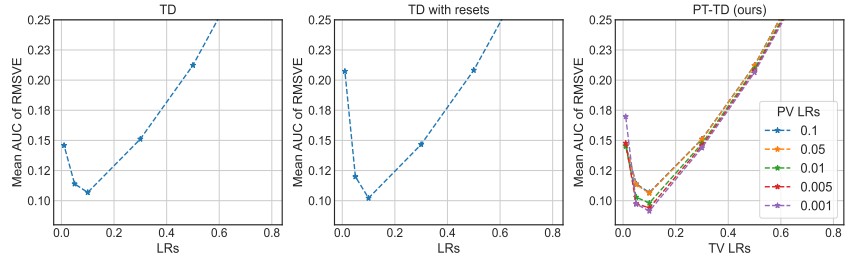

Figure C.10: Mean AUC of RMSVE against learning rate for various algorithms.

For tabular and linear grid tasks, we select best hyperparameters (learning rates) for each algorithm by choosing those values which results in the lowest area under the error curves as shown in Table C.6, Figures C.9, C.10, C.11. The hyperparameter search and the final reporting is performed by averaging the results on 30 different seeds. For the deep RL task with discrete grid, we choose the best learning rate from the values shown in Table C.7 and the corresponding U curves are shown in C.12. For the minigrid policy evaluation task, we search for the best learning rates from the values shown in Table C.8 and the corresponding U curves are shown in Fig. C.13.

For the tabular control experiment, the learning rates used to search for the best value is listed in Table C.9 and the mean returns for various choices of these values are plotted in Fig. C.14 for various algorithms. Similar table for the minigrid task (deep control) is shown in Table C.10 and the mean returns for various choices of these values is shown in Fig. C.15. We use 30 seeds to pick the best learning rate for the tabular task but use 3 seeds to select the learning rates for the minigrid task.

The learning rate tables and the corresponding performance for the full continual RL experiments are shown in Fig. C.16 and Tables C.11, C.12, C.17, C.18, C.19, and C.20. We use 30 seeds to find the best hyperparmeters for the tabular task, but we use 3 seeds for JBW experiment and MinAtar.

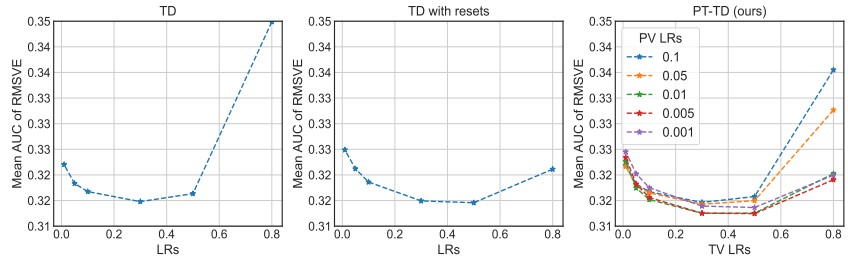

Figure C.11: Mean AUC of RMSVE against learning rate for various algorithms.

| Algorithm | LRs |
|---|---|
| Deep TD (without reset) | {1e-1, 3e-2, 1e-2, 3e-3, 1e-3} |
| Deep TD (with reset) | {1e-1, 3e-2, 1e-2, 3e-3, 1e-3} |
| Deep PT-TD | PV: {1e-3, 3e-4, 1e-4, 3e-5} |
| | TV: {1e-2, 5e-3, 3e-3, 1e-3} |

Table C.7: LRs considered to pick the best one for the Deep prediction experiment with discrete grid.

We find the best learning rate for each $k, \lambda$ pair by searching over the learning rates listed in Table C.11.

**Observations:** We see that our algorithm performs better than their vanilla counterparts on all the problems we tested for a broad range of hyperparameters. There are a handful of $(\overline{\alpha}, \alpha)$ pairs that results in a better performance in each setting, which indicates that our method is not sensitive to hyperparameter selection. We also observe that in all the cases $\overline{\alpha}$ is smaller than $\alpha$, which fits into our intuition. Because, the permanent value function has to capture regularities in the value function estimate, it needs to update slowly, whereas, the transient value function should be updated quickly to facilitate fast adaptation. The sensitivity of two new hyperparameters: $\lambda, k$ for the fully continual RL setting is analyzed in the main paper (see Sec. 6.1) where we concluded that our algorithm is robust to these values.

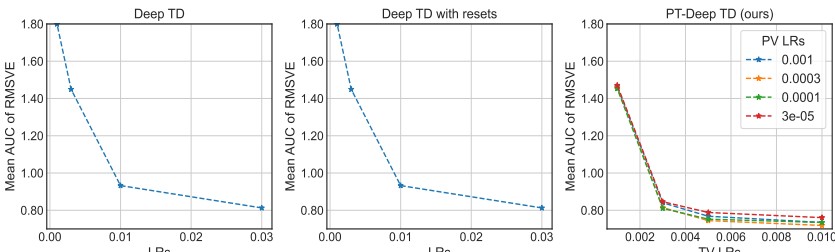

Figure C.12: Mean AUC of RMSVE against learning rate for various algorithms on image grid.

| Algorithm | LRs |
|---|---|
| TD (without reset) | {1e-1, 3e-2, 1e-2, 3e-3, 1e-3} |
| TD (with reset) | {1e-1, 3e-2, 1e-2, 3e-3, 1e-3} |
| permanent value function | {3e-2, 1e-2, 3e-3, 1e-3, 3e-4} |
| transient value function | {3e-3, 1e-3, 3e-4, 1e-4, 3e-5, 1e-5, 3e-6, 1e-6} |

Table C.8: Search space of learning rate for the policy evaluation minigrid task.

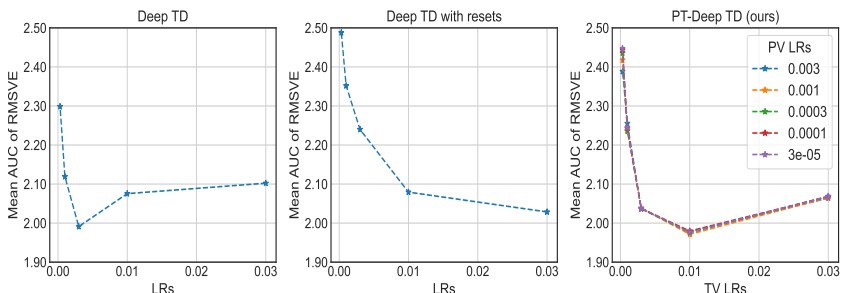

Figure C.13: Mean AUC of RMSVE against learning rate for various algorithms on minigrid grid.

| Algorithm | LRs |
|---|---|
| Q-learning | {8e-1, 5e-1, 1e-1, 5e-2, 1e-2, 5e-3, 1e-3} |
| Q-learning (with reset) | {8e-1, 5e-1, 1e-1, 5e-2, 1e-2, 5e-3, 1e-3} |
| Permanent Value Function | {8e-1, 5e-1, 3e-1, 1e-1, 5e-2, 1e-2, 5e-3, 1e-3} |
| Transient Value Function | {8e-1, 5e-1, 3e-1, 1e-1, 5e-2, 1e-2} |

Table C.9: Search space of learning rates for the tabular control task.

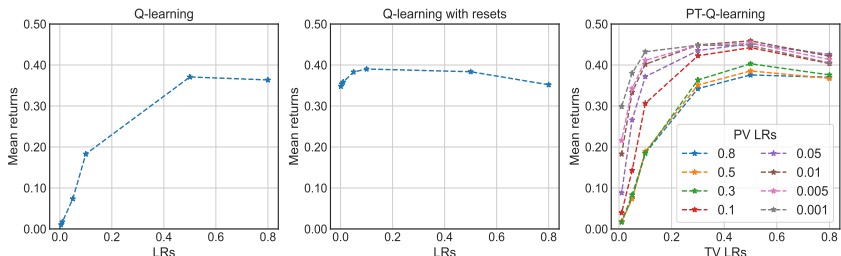

Figure C.14: Mean returns for various learning rates for the tabular control task.

| Algorithm | LRs |
|---|---|
| Q-learning | {3e-3, 1e-3, 3e-4, 1e-4, 3e-5} |
| Q-learning (with reset) | {1e-2, 3e-3, 1e-3, 3e-4, 1e-4} |
| Permanent Value Function | {1e-5, 3e-6, 1e-6, 3e-7} |
| Transient Value Function | {3e-4, 1e-4, 3e-5, 1e-5} |

Table C.10: Search space of learning rates.

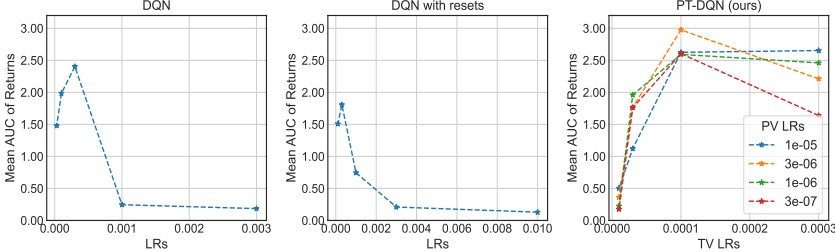

Figure C.15: Mean returns for various learning rates for the Minigrid control task.

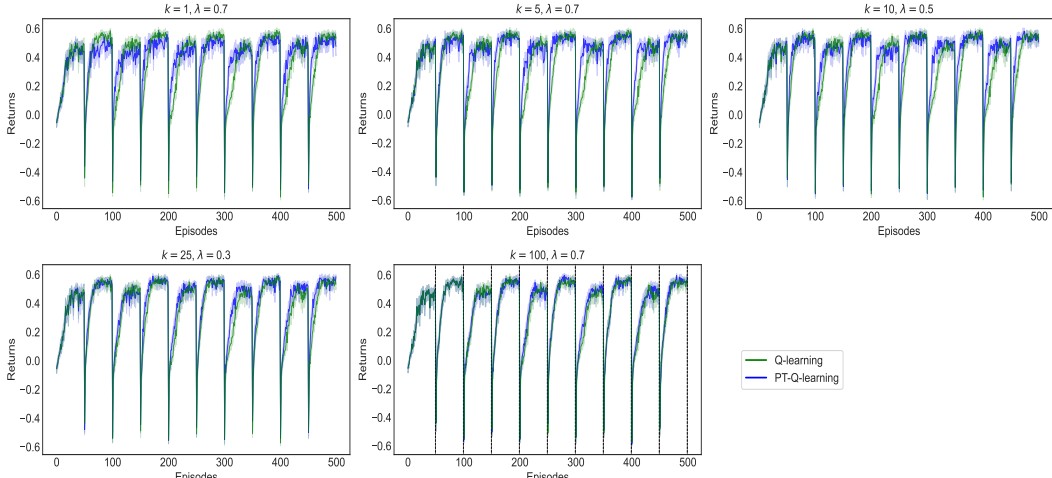

Figure C.16: Performance plot of PT-Q-learning against Q-learning for $k, \lambda$ pairs

| Algorithm | LRs |
|---|---|
| Q-learning | {8e-1, 5e-1, 1e-1, 5e-2, 1e-2, 5e-3, 1e-3} |
| Permanent Value Function (0.5x) | {1e-4, 1e-5, 1e-6, 1e-7, 1e-8} |
| Transient Value Function (0.5x) | {1e-2, 1e-3, 1e-4, 1e-5} |

Table C.11: Search space of learning rates for the main JBW experiment.

| Algorithm | LRs |
|---|---|
| Q-learning | {1e-2, 1e-3, 1e-4, 1e-5} |
| Permanent Value Function | {1e-4, 1e-5, 1e-6, 1e-7} |
| Transient Value Function | {1e-2, 1e-3, 1e-4, 1e-5} |

Table C.12: Search space of learning rates for the main JBW ablation experiment.

| TV-LR → PV-LR ↓ | $\lambda$=0.55 | | | | $\lambda$=0.75 | | | | $\lambda$=0.95 | | | |
|---|---|---|---|---|---|---|---|---|---|---|---|---|
| | 1e-2 | 1e-3 | 1e-4 | 1e-5 | 1e-2 | 1e-3 | 1e-4 | 1e-5 | 1e-2 | 1e-3 | 1e-4 | 1e-5 |
| 1e-4 | 0.031 | **0.223** | 0.126 | 0.103 | 0.035 | **0.257** | 0.154 | 0.154 | 0.038 | **0.217** | **0.199** | 0.159 |
| 1e-5 | 0.031 | **0.252** | 0.071 | 0.051 | 0.030 | **0.261** | 0.077 | 0.068 | 0.040 | **0.220** | **0.201** | 0.096 |
| 1e-6 | 0.041 | **0.242** | 0.070 | 0.061 | 0.034 | **0.260** | 0.090 | 0.083 | 0.030 | **0.234** | **0.207** | 0.084 |
| 1e-7 | 0.036 | **0.256** | 0.056 | 0.067 | 0.040 | **0.268** | 0.062 | 0.070 | 0.028 | **0.227** | **0.216** | 0.124 |
| 1e-8 | 0.031 | **0.247** | 0.051 | 0.069 | 0.041 | **0.261** | 0.069 | 0.069 | 0.037 | **0.232** | **0.213** | 0.080 |

Table C.13: JBW: Reward per step for various hyperparameters for PT-DQN. We highlight the values that is larger than DQN's best performance.

| LRs | 1e-2 | 1e-3 | 1e-4 | 1e-5 |
|---|---|---|---|---|
| Returns | 0.03 | 0.127 | **0.192** | 0.093 |

Table C.14: JBW: Reward per step for various hyperparameters for DQN.

| LRs | 1e-2 | 1e-3 | 1e-4 | 1e-5 |
|---|---|---|---|---|
| Returns | 0.037 | 0.135 | **0.247** | 0.245 |

Table C.15: JBW: Reward per step for various hyperparameters for DQN-multi-head.

| LRs | 1e-2 | 1e-3 | 1e-4 | 1e-5 |
|---|---|---|---|---|
| Returns | 0.025 | 0.055 | **0.055** | 0.031 |

Table C.16: JBW: Reward per step for various hyperparameters for DQN with large experience replay.

| TV-LR → PV-LR ↓ | $\lambda$=0.55 | | | $\lambda$=0.75 | | | $\lambda$=0.95 | | |
|---|---|---|---|---|---|---|---|---|---|
| | 1e-3 | 1e-4 | 1e-5 | 1e-3 | 1e-4 | 1e-5 | 1e-3 | 1e-4 | 1e-5 |
| 1e-7 | **14.41** | **17.38** | 7.89 | **13.87** | **19.07** | 10.55 | 9.50 | **16.12** | 12.16 |
| 1e-8 | **15.29** | **17.35** | 7.79 | **14.02** | **20.54** | 10.72 | 9.79 | **16.71** | 12.48 |
| 1e-9 | **15.54** | **17.51** | 8.02 | 12.63 | **19.54** | 10.90 | 8.57 | **16.48** | 11.76 |

Table C.17: MinAtar: AUC of the average return (in the previous 100 episodes) per step for various hyperparameters for PT-DQN. We highlight the values that is larger than DQN's best performance.

| LRs | 1e-3 | 1e-4 | 1e-5 | 1e-6 |
|---|---|---|---|---|
| Returns | 7.10 | 12.90 | **13.14** | 5.90 |

Table C.18: MinAtar: AUC of the average return (in the previous 100 episodes) per step for various hyperparameters for DQN.

| LRs | 1e-3 | 1e-4 | 1e-5 | 1e-6 |
|---|---|---|---|---|
| Returns | 8.72 | 14.93 | **15.68** | 8.54 |

Table C.19: MinAtar: AUC of the average return (in the previous 100 episodes) per step for various hyperparameters for DQN-multi-head.

| LRs | 1e-3 | 1e-4 | 1e-5 | 1e-6 |
|---|---|---|---|---|
| Returns | 6.78 | **11.44** | 10.70 | 4.56 |

Table C.20: MinAtar: AUC of the average return (in the previous 100 episodes) per step for various hyperparameters for DQN with large experience replay.

