# OpenReview forum: "Prediction and Control in Continual Reinforcement Learning"
_NeurIPS.cc/2023/Conference — NeurIPS 2023 poster_

### Official Review · Reviewer_mEyq · 2023-06-28

**Soundness:** 4 excellent
**Presentation:** 3 good
**Contribution:** 2 fair
**Rating:** 7
**Confidence:** 3

**Summary:**

In this work, the author proposes a new continual reinforcement learning method, the PT-TD, which consists of two value networks, one learns the transient memories and the other learns the permanent memories. The author proves that in the semi-continual setting, the PT-TD algorithm will get better performance and lower forgetting, and the same result can also be found in the continual reinforcement learning setting.

**Strengths:**

The algorithm of this article is well-proven and well-demonstrated. The author not only proves the PT-TD algorithm in a mathematical way but also gives out detailed and rigorous experiments. Also, the article is really well-written and easy to understand. All of the hyperparameters are studied. Meanwhile, it is appreciated that the author discussed semi-continual reinforcement learning and continual reinforcement learning separately.

**Weaknesses:**

First, even though the proof is very well, the algorithm itself is not very groundbreaking, or, the result is not very rewarding. Besides, the environments of the experiments are simple.

**Questions:**

I think the formulation of the Theorem 8 is a little confusing. Maybe you can give out explicitly which network is using (P or T).

Can your algorithm generalize to continuous action space? How should I learn the actor network at this time?

I notice that in the continual reinforcement learning setting, the learning rate of T network is always higher than it of P network. According to [1], if it is the difference in the learning rate affect good performance?

[1]Arani, E. et al. “Learning Fast, Learning Slow: A General Continual Learning Method based on Complementary Learning System.” ArXiv 2021

---

> ### Author Rebuttal · Authors · 2023-08-10
>
> Thank you for your review and positive comments on our paper. Please find our answers to your questions below:
>
> > 1. I think the formulation of the Theorem 8 is a little confusing. Maybe you can give out explicitly which network it is using (P or T).
>
> We will update the statement in the main paper using the one provided in the appendix (Theorem B.3.4) to make it clear. Intuitively, theorem 8 states that the prediction errors on a new task are lower for our method compared with TD-learning. The theorem follows Algorithm 1 where we fix the parameters of the permanent network and update only the transient network using online data. The overall estimate is the sum of the predictions of the two networks.
>
> > Can your algorithm generalize to continuous action space? How should I learn the actor network at this time?
>
> Q-learning with continuous actions can be implemented by passing the observation and action as input to estimate the Q-values. The permanent and transient value functions could then be used as in the paper. Alternatively, the action space can be discretized to learn Q-values using the network architecture described in the paper.
>
> Another possibility is to pursue policy gradient approaches. There is a straightforward way to extend the work to actor-critic methods, by simply using the proposed value function estimates to compute the gradient of the policy (represented as a separate approximator). Another possibility is to have a "permanent" policy, which is corrected by a transient component to adapt to the task at hand. Both would be worth exploring in future work and we will reflect this in the discussion.
>
> > I notice that in the continual reinforcement learning setting, the learning rate of T network is always higher than it of P network. According to [1], if it is the difference in the learning rate affect good performance?
>
> The difference in learning rates is part of the answer, but other aspects of the algorithm play a bigger role:
>
> (a) The permanent and the transient memories are updated at different timescales. The permanent network is updated at a slower timescale (every $k$ steps or at the end of the task). The transient memory is updated at a faster timescale (every timestep), therefore, it absorbs errors immediately (fast);
>
> (b) The update rules of permanent and transient memories are different. The permanent memory is updated to capture some part of the value function from all tasks that the agent has seen in its lifetime (generalization), which is a slow process. The transient memory adapts to a specific situation, which is a fast process.
>
> The fast and slow interplay on various levels results in good performance on non-stationary problems.
>
> > The algorithm itself is not very groundbreaking, or, the result is not very rewarding. Besides, the environments in the experiments are simple.
>
> We deliberately tried to keep our approach as simple as possible and build on top of standard TD-learning and Q-learning algorithms. This is desirable as TD-learning and Q-learning have demonstrated generality and scalability on a broad range of problems over time, despite (or perhaps because) of being simple.
>
> Regarding the experiments, we used domains that are typical in semi-continual RL and continual RL (Minigrid, JellyBeanWorld, and MinAtar). The JellyBeanWorld and MinAtar experiments in particular, while not being "real applications", do test scaling.
> The type of experiments we ran is consistent with other papers in the field, but the tasks we used are harder than in earlier works. For instance, [1] uses catcher and flappy bird with a curriculum of tasks, while the non-stationarity in our setup is much more severe. [2] used only one continual RL problem - slippery ant. Any further concrete suggestions for experiments would be useful.
>
> *References:*
>
> [1] Riemer, et al. "Learning to learn without forgetting by maximizing transfer and minimizing interference." arXiv preprint arXiv:1810.11910 (2018).
>
> [2] Dohare, et. al. "Continual backprop: Stochastic gradient descent with persistent randomness." arXiv preprint arXiv:2108.06325 (2021).

---

> > ### Comment · Reviewer_mEyq · 2023-08-14
> >
> > Thanks to the author for the rebuttal. Just to further refine this paper, I propose to add experiment results of the setting that both P and T networks use the same learning rate.

---

> > > ### Author Response · Authors · 2023-08-16
> > >
> > > Thank you for your suggestion. We tested our method on a broad range of learning rate values including the cases where they are the same for both the P and T networks. We picked the combination that resulted in the smallest error (or the largest reward) for each experiment. We included a detailed report on the hyperparameter selection in the appendix (Section C4). In summary: (a) Our approach is less sensitive to learning rate choices when compared with its vanilla counterparts; (b) The performance degraded slightly when we used large learning rate values to train the permanent memory (this is expected because we want the permanent memory to take small steps to allow slow knowledge consolidation). We will elucidate our observations in the main paper based on our discussion, and include a reference to the suggested paper.

---

### Official Review · Reviewer_zZwF · 2023-07-05

**Soundness:** 3 good
**Presentation:** 4 excellent
**Contribution:** 3 good
**Rating:** 7
**Confidence:** 3

**Summary:**

This paper proposes new algorithms for prediction and control in continual reinforcement learning. The new algorithms use permanent and temporary memories. The permanent memory captures a general value function for all the tasks, while the transient memory is used to adapt to the current tasks continually. The paper provides theoretical results for the prediction setting with a tabular learner. Additionally, the paper provides empirical results showing that their method outperforms vanilla TD-based algorithms in various environments using different types of function approximators.

**Strengths:**

The paper directly tackles the continual reinforcement learning setting.
This is a challenging problem setting with few general algorithms and evaluation environments.
The paper provides prediction and control algorithms for the continual RL setting.
Theoretical proofs of convergence are provided for the prediction setting with a tabular learner.
The extensive and sound empirical investigation shows that the proposed method can generally outperform existing methods in various environments and with different various function approximations, including Deep Networks.

**Weaknesses:**

The paper does not have any major weaknesses. I did not check the proofs, so I can not comment on the correctness of the proofs.
However, there are a few minor issues that can be fixed to improve the paper.
1. I'm confused about Equation 5, specifically line 112. Does it mean the "gradient" or the "semi-gradient" of the transient memory?
2. What are the confidence intervals being reported? Line 222 says it is the 90% confidence interval, but is the bootstrapped confidence interval or confidence interval based on the standard error? And if it is based on the standard error, how do you know it is the 90% confidence interval?
3. The continual deep RL algorithms used in the paper use standard deep RL tools like Experience replay and target networks. However, the exact use of these tools is not fully explained in the paper (or the appendix). This creates a lot of questions; for example, are there a total of three networks, target, transient and permanent? I suggest that the authors include a section in the appendix that gives the pseudocode of the PT-DQN algorithm. The lack of details about the PT-DQN algorithm makes it hard to replicate the results given in the paper.


**Questions:**

1. Line 88 says, "... multi-task RL methods ..." What is multi-task RL, and how is it different from continual RL? What limits the use of multi-task RL methods in continual RL? The second question should be answered in the main text.
2. Figure 4d shows the performance of different versions of DQN  on Minigrid. Why is the performance of all the methods, particularly DQN(reset), so much worse during the second task? If the DQN is being reset at each task, it should perform similarly on Episodes 500 and 1000. Is this because the first goal state in all the runs is the same?
3. What is PT-DQN-0.5x in Figure 5? 0.5x should be described in the main paper.
4. Lines 733-742 describe the training details for the deep RL experiments. However, many of these choices seem arbitrary, or the paper does not explain the reasoning behind these choices. Why are the conv layers not updated when updating the permanent memory? Why isn't there a fully independent permanent value network? Wouldn't that be closer to the algorithms presented in the paper?
5. Lines 803-805 say that the permanent memory was trained with SGD while the rest of the network was trained with Adam. Why this specific choice? I'm guessing the authors also tried Adam to update the permanent memory. Why didn't Adam work for permanent memory?
6. The plots (like Figure C.13) showing the sensitivity curve will be better with a log scale for the x-axis.
7. Why isn't there any effect of PM LR on the performance in the minigrid environment (Figure C.13c)?

**Limitations:**

One limitation of this work is that it only proposes methods for value-based RL. It is unclear how the idea of permanent and transient memory can be extended to policy-gradient based RL. A discussion on this direction of future work will be helpful.

---

> ### Author Rebuttal · Authors · 2023-08-10
>
> Thank you for your thorough review and positive comments on our paper. Please find our answers to your questions below:
>
> > I'm confused about Equation 5, specifically line 112. Does it mean the "gradient" or the "semi-gradient" of the transient memory?
>
> This is a semi-gradient update rule, in the same sense used to describe TD-learning, as the gradient is with respect to the current estimate only and not the target. Since the permanent and the transient memories are independently parameterized, the semi-gradient ends up being just $\nabla_w V_w^T(S_t)$.
>
> > What are the confidence intervals being reported?
>
> We reported standard distribution confidence intervals with a z-score of 1.645 and a population size of 30 (seeds). We used $\mu \pm z \frac{\sigma}{\sqrt{n}}$ to represent the shaded region, where $\mu$ is the mean performance, $\sigma$ is the standard deviation of the performance over the population, and $n$ is the number of seeds.
>
> > [...] pseudocode of the PT-DQN algorithm. [...]
>
> We value reproducibility and we will release the code for all the experiments so others can replicate our results. We will also expand the appendix to include the pseudocode of the continual deep RL algorithms along with the details of the deep RL tools used in experiments to provide more clarity.
>
> > [...] What is multi-task RL [...]
>
> We define the multi-task setting in lines 67-69, and we will expand this description. We use the term multi-task RL to mean that the agent is confronted with a distribution of RL problems (tasks) and that during learning, the agent is aware of the task ID and/or of whether the task has changed since its previous episode. In some cases, the agent can sample tasks from a distribution, thereby controlling the data it observes to a certain extent. Methods in multi-task RL are designed to make use of this additional information, which is not available in continual RL (for example, see DQN-multi-head baseline Section 6). These methods have certain limitations as described in lines 32-35 when applied to continual RL: they require the agent to first identify the task and update appropriate parameters, which limits the number of tasks and the kind of non-stationarity (only piece-wise) that can be considered.
>
> > [...] so much worse during the second task? [...]
>
> The first and second goals and the task order are the same for all baselines and for all runs. The DQN agent took longer to solve the second task when we trained it on each task separately (see performance between 500-1000 episodes, and 1500-2000 episodes). We suspect this is because the first goal is closer to the start state (it is the goal on the left), and therefore easier for the agent to discover during random exploration. We are unsure why the gap is as large as it is, and we will investigate this further. Note that the other methods, which are not reset, can leverage some of the previous learning, and they adapt faster to this goal than the reset baseline.
>
> > What is PT-DQN-0.5x in Figure 5?
>
> We will add details in the main paper. It is a version in which the permanent and the transient memory networks have respectively half the number of parameters used by the DQN network, to ensure the total number of parameters across all baselines is the same. Strictly speaking, DQN-based baselines use more parameters compared with PT-DQN-0.5x because their target networks are 2x larger than the target network of the transient memory.
>
> > [...] the paper does not explain the reasoning behind these choices. [...]
>
> Since the task boundaries are observed, we chose to train our method in a similar manner to multi-task RL methods (for example, see DQN-multi-head baseline). We only update the corresponding head to learn the permanent value because the features are already learned when updating the transient memory. We will clarify this in the text.
>
> > Why isn't there a fully independent permanent value network? [...]
>
> Indeed, we parameterize the permanent and transient memory networks independently in the continual RL experiments (see Section 6) to align with the algorithm introduced in the paper. In general, one may in fact want different network architectures for these, not just different parameters, but this goes beyond the scope of this paper.
>
> > [...] the permanent memory was trained with SGD while the rest of the network was trained with Adam. [...]
>
> The permanent memory network should take a small step during each update in order to consolidate the long-term value function information. SGD's updates are more aligned with this intuition, which is why we preferred it. We also tested the Adam optimizer initially; the performance was poor with default hyperparameters. We expect Adam's performance could match or exceed SGD's when all its hyperparameters are fine-tuned, but we did not try that.
>
> We used the standard DQN training procedure (experience replay, Adam optimizer, target network) for training the DQN baselines, and the transient memory network.
>
> > The plots [...] with a log scale for the x-axis.
>
> We will incorporate this suggestion in the final version of the appendix.
>
> > Why isn't there any effect of PM LR [...]
>
> In general, we observed a good performance for a wide range of PM LRs in various environments. In minigrid, the difference seems to be small compared with other environments; perhaps learning permanent values is easier in that environment.
>
> > Extension to policy-based RL
>
> While the paper focuses on value-based RL, there is a straightforward way to extend the work to actor-critic methods, by simply using the proposed value function estimates to compute the gradient of the policy (represented as a separate approximator). Another possibility is to have a "permanent" policy, which is corrected by a transient component to adapt to the task at hand. Both would be worth exploring in future work and we will reflect this in the discussion.
>
> Thank you again for all your questions and suggestions to improve the paper.

---

> > ### Comment · Reviewer_zZwF · 2023-08-16
> >
> > I thank the authors for answering my questions. I maintain my score and recommendation for acceptance.

---

### Official Review · Reviewer_27Kd · 2023-07-05

**Soundness:** 3 good
**Presentation:** 3 good
**Contribution:** 3 good
**Rating:** 6
**Confidence:** 4

**Summary:**

The paper proposes a method to improve continual learning in reinforcement learning with non-stationary dynamics. It proposes to maintain two value functions: one that tracks information about the current task (transient) and one that maintains a general knowledge of the task distribution (permanent) in order to better tackle the non-stationarity of the environment.


**Strengths:**

1. The general idea is simple in terms of intuition and implementation. I like the idea of maintaining two different value functions operating different time-scales.
2. Assuming soundness (see below), the experiments well-done and are insightful.

**Weaknesses:**

While the overall approach is intuitive, the specifics dont seem very intuitive to me and there is not much information given about it. For example, Eq 4 (see questions below) is not intuitive to me on how it was derived. Similarly, I have other questions.

**Questions:**

1. What do the authors mean when they say prior work “reflected the nature of that task” in line 41?
2. What is the intuition for the update in Eq 4? Why does the update to the parameters of the P get scaled by V^PT? Similarly, what is the intuition for Eq 5? Why is \delta purely in terms of the overall update, PT?
3. The definition in Eqn 2 and 3 is that both memories are equally weighted and are just a sum. I am curious what the authors think about this equal weighting and the consequences of it?
4. It is a little unclear to me why only the initialization condition is sufficient for Theorem 1 to hold?
5. Shouldn’t the equation in Theorem 2 be what is in Equation 5 i.e. PT instead of P?
6. It may be worthwhile to see how Continual backprop [1] can be incorporated into the proposed method.

[1] Continual Backprop: Stochastic Gradient Descent with Persistent Randomness. Dohare et al.


**Limitations:**

See above questions

---

> ### Author Rebuttal · Authors · 2023-08-10
>
> Thank you for your review. We answer your questions below, please let us know if further clarification is needed.
>
> > What do the authors mean when they say prior work “reflected the nature of that task” in line 41?
>
> We meant that the method presented in Silver et al [1] leverages domain knowledge about the game of Go, which is the task they consider, to derive the planning update rule.
>
> > What is the intuition for the update in Eq 4? Why does the update to the parameters of the P get scaled by $V^{PT}$? Similarly, what is the intuition for Eq 5? Why is $\delta$ purely in terms of the overall update, PT?
>
> We want the overall estimate, $V^{PT}$, to approximate well the state value function. In order to provide a good starting point for any given task, the permanent value function should "mix" value estimates from all tasks. This is achieved by taking a step towards the new estimate of $V^{PT}$ at a slow pace (either at the end of the task or every $k$ steps). This is Eq. (4), which is basically similar to a supervised learning update with $V^{PT}$ as the target. Theorems 6 and 7 show that this update has favourable theoretical properties.
>
> Since some part of the value is already captured by $V^P$, $V^T$ should compute only further corrections from it, which is implemented in Eq.(5). We agree that the use of the overall TD-error in this update is a bit surprising, but this falls out of the math. Theorems 2 and 3 prove that this is the correct update given $V^P$.
>
> > The definition in Eqn 2 and 3 is that both memories are equally weighted and are just a sum. I am curious what the authors think about this equal weighting and the consequences of it?
>
> Our choice was meant to be as simple as possible and was inspired by an earlier paper in model-based RL [1] as discussed in Sections 1 and 2. In general, emphasizing $V^P$ increases bias towards the past value functions observed, which can reduce variance. Emphasizing $V^T$ would have the opposite effect. Other choices, like linear or non-linear combinations, could be explored in future work. It may also be possible to meta-learn how to combine these estimates, or to use side information, such as a task description, to condition a more flexible combination scheme. However, in continual learning, it may be difficult to make good choices unless some information about future task distribution is available.
>
> > It is a little unclear to me why only the initialization condition is sufficient for Theorem 1 to hold?
>
> The proof (included in the Appendix) contains the details. When the initial condition is met along with step-size and data stream requirements, the predictions of our algorithm match those learned using TD-learning. This is not surprising because the transient memory updates use TD-learning, but the semi-gradient is with respect to the transient component only instead of the overall estimate $V^{PT}$.
>
> > Shouldn't the equation in Theorem 2 be what is in Equation 5 i.e. PT instead of P?
>
> Indeed, the Bellman operator in Theorem 2 is obtained by using the target in Eq. 5. $\gamma P_{\pi} V^P$ and $\gamma P_{\pi} V^T$ add up to $\gamma P_{\pi} V^{PT}$ by definition.
>
> > It may be worthwhile to see how Continual backprop can be incorporated into the proposed method.
>
> Our method performs a trade-off between stability and plasticity in the value function, that is agnostic to the nature of the function approximator. Continual backprop is an optimizer explicitly designed to promote neuron-level plasticity in neural networks. In principle, if the value functions are implemented using neural networks, this optimizer can be used to improve plasticity further. We expect that using continual backprop would be especially useful to update the permanent memory network as it runs the risk of losing neuron-level plasticity when aggregating the value estimates from several tasks. This is an interesting direction to explore.
>
> We will edit the paper to incorporate these clarifications as well, let us know if you have further comments or questions.
>
> *References:*
>
> [1] David Silver, Richard S Sutton, and Martin Müller. Sample-based learning and search with permanent and transient memories. In Proceedings of the 25th international conference on Machine learning, pages 968–975, 2008.

---

> > ### Comment · Reviewer_27Kd · 2023-08-11
> >
> > Thank you to authors for clarifying my questions. I will update my score.

---

### Official Review · Reviewer_mFKc · 2023-07-06

**Soundness:** 2 fair
**Presentation:** 3 good
**Contribution:** 2 fair
**Rating:** 4
**Confidence:** 3

**Summary:**

This work proposes a reinforcement learning method for prediction and control. The value function in reinforcement learning is decomposed into two terms: a permanent value function, and a transient value function. The permanent function contains the long term knowledge of the interplay between the agent and the environment, and the transient value function absorbs the quick information variations. This work uses temporal difference learning for updating the estimate of the value function estimation. The learning process is related to the complementary learning systems from neuroscience. Prediction experiments are performed for 5x5 discrete grid. The agent moves from the center and selects an action the goal states.

**Strengths:**

It is of great interest to extract valuable information from stream data. The development of organic transistor featuring a selective transition from short-term to long-term plasticity of the biological synapse. Catastrophe forgetting is a major challenge for machine learning. A possible solution for solving such a problem is very attractive. It is closely related to meta learning.

**Weaknesses:**

The idea for separating long term and transient information is intuitive. It has been used in many machine learning algorithms such as LSTM.  It is not clear if a long term function is always available, especially for complicated interplay between an agent and its environment. The separation of long term and transient can be dynamic.

The settings for the experiments are very simple. It does not demonstrate the possibilities for real applications.


**Questions:**

In the experiments, the proposed method is compared with DQN. Is it possible to compare other competitive algorithms?

**Limitations:**

The limitations of this work are not explicitly given.

---

> ### Author Rebuttal · Authors · 2023-08-10
>
> Thank you for the review. We address the main points below.
>
> > The settings for the experiments are very simple.
>
> We designed the experiments in the paper to cover several aspects that we deemed important: environment non-stationarity, different function approximators (tabular, linear, non-linear), generalization ability, and ability to deal with large observation spaces (such as RGB images). We used domains that are typical in semi-continual RL and continual RL (Minigrid, JellyBeanWorld, and MinAtar). The JellyBeanWorld and MinAtar experiments in particular, while not being "real applications", do test scaling.
>
> The type of experiments we ran is consistent with other papers in the field, but the tasks we used are harder than in earlier works. For instance, [1] uses catcher and flappy bird with a curriculum of tasks, while the non-stationarity in our setup is much more severe. [2] used only one continual RL problem - slippery ant. We test our algorithm on a diverse set of problems and function approximators.
>
> If you consider these experiments insufficient, it would be very useful to know what additional aspects of our approach you would have found useful to test, or what environments suitable for continual learning you would suggest. Concrete suggestions of environments and/or experiments that can be carried out in an academic environment would be helpful.
>
> > The proposed method is compared with DQN. Is it possible to compare other competitive algorithms?
>
> Our approach to choosing baselines is to compare against the algorithms on which we are building. We used TD learning as the baseline for policy evaluation because our work builds on TD, and the transient memory is updated using a TD-like rule. Q-learning is used in the control experiments for the same reason. This approach is similar to other research papers like [1] and [2]. In [1], DQN with experience replay is compared against the proposed DQN with meta-experience replay. [2] uses PPO with backprop as a baseline for continual backprop.
>
> **Also, in the continual RL experiments in Section 6, we introduced 2 additional competitive baselines:** (a) DQN paired with a large experience replay buffer &mdash; an approach proposed recently as a viable option for continual RL [3]; (b) DQN with a common trunk to learn features and separate heads to learn Q-values for the different tasks &mdash; a common approach when task boundaries are observable [4].
>
> Note that our approach could be wrapped around other RL methods that leverage value functions, in which case it would make sense to compare against those methods.
>
> > The limitations of this work are not explicitly given.
>
> As reviewer zZeF pointed out, our work is limited to methods which leverage a value function. Also, our theoretical analysis is limited to piece-wise non-stationarity, where the environment is stable for some period of time. An interesting direction would be to extend our approach to other types of non-stationarity, such as continuous wear and tear of robotic plants. We will add a paragraph in the discussion section to address this.
>
> > The idea for separating long term and transient information is intuitive. [...] The separation of long term and transient can be dynamic
>
> Intuitive ideas are often the ones that stand the test of time. In our case, we build on CLS neuroscience intuitions, and we deliberately try to do so in the simplest way possible. We agree that different ways of combining permanent and transient memories, perhaps conditioned on a task description or through meta-learning could be interesting to explore in the future. But we were very pleased that the simplest approach that we could conceive actually leads to strong results in challenging continual RL tasks.
>
> *References:*
>
> [1] Riemer, et al. "Learning to learn without forgetting by maximizing transfer and minimizing interference." arXiv preprint arXiv:1810.11910 (2018).
>
> [2] Dohare, et. al. "Continual backprop: Stochastic gradient descent with persistent randomness." arXiv preprint arXiv:2108.06325 (2021).
>
> [3] Massimo, et. al.. Task-agnostic continual reinforcement learning: In praise of a simple baseline. arXiv preprint arXiv:2205.14495, (2022).
>
> [4] Samuel, et. al. Same state, different task: Continual reinforcement learning without interference. AAAI Conference on Artificial Intelligence, (2022).

---

### Author Rebuttal · Authors · 2023-08-10

Thank you for your time and useful suggestions for improving our paper. We appreciate the positive feedback on the ideas, their connections to complementary learning systems and the theoretical analysis. We address specific questions below and would be happy to provide any further clarifications.

---

### Decision · Program_Chairs · 2023-09-21

**Decision:**

Accept (poster)

**Comment:**

This paper focuses on continual RL and proposes to decompose the value function into two parts: persistent and transient memory. The proposal is backed up by theory and the authors also demonstrate the use of the proposed algorithm in several empirical settings.

The authors addressed most of the reviewers' concerns and reviewers also raised their scores. There were some complaints about the need for more large-scale experiments. But I think the work is interesting enough and worth publishing. I recommend an acceptance.

I encourage the authors to

1. Add a detailed limitation section to the paper.
2. A detailed pseudocode of the algorithm in the appendix.
3. Incorporate all the clarifications made during the rebuttal period.